# *PH13* improves soybean shade traits and enhances yield for high-density planting at high latitudes

Chao Qin[1,17], Ying-hui Li[1,17], Delin Li[1,17], Xueru Zhang[2], Lingping Kong[3], Yonggang Zhou[4], Xiangguang Lyu[1], Ronghuan Ji[1], Xiuzhi Wei[1], Qican Cheng[1], Zhiwei Jia[5], Xiaojiao Li[5], Qiang Wang[6], Yueqiang Wang[7], Wen Huang[8], Chunyan Yang[9], Like Liu[10], Xing Wang[11], Guangnan Xing[12], Guoyu Hu[13], Zhihui Shan[14], Ruizhen Wang[15], Haiyan Li[4], Hongyu Li[1], Tao Zhao[1], Jun Liu[1], Yuping Lu[5], Xiping Hu[16], Fanjiang Kong[3] ✉, Li-juan Qiu[1] ✉ & Bin Liu[1] ✉

Shading in combination with extended photoperiods can cause exaggerated stem elongation (ESE) in soybean, leading to lodging and reduced yields when planted at high-density in high-latitude regions. However, the genetic basis of plant height in adaptation to these regions remains unclear. Here, through a genome-wide association study, we identify a plant height regulating gene on chromosome 13 (*PH13*) encoding a WD40 protein with three main haplotypes in natural populations. We find that an insertion of a *Ty1/Copia*-like retro-transposon in the haplotype 3 leads to a truncated PH13[H3] with reduced interaction with GmCOP1s, resulting in accumulation of STF1/2, and reduced plant height. In addition, *PH13[H3]* allele has been strongly selected for genetic improvement at high latitudes. Deletion of both *PH13* and its paralogue *PHP* can prevent shade-induced ESE and allow high-density planting. This study provides insights into the mechanism of shade-resistance and offers potential solutions for breeding high-yielding soybean cultivar for high-latitude regions.

Soybean (*Glycine max* (L.) Merr.) is an economically important crop, accounting for 59% of the oilseed production and providing 70% of the plant protein for human and animal consumption worldwide (Soy-Stats, 2023)[1]. To meet the demands of an ever-growing population and to continuously improve the living standards, it is estimated that the global soybean yield must be doubled by 2050[2,3]. Soybean is a short-day plant (SDP) that originated in the temperate regions of China (between 32 and 40°N)[4]. However, traditional soybean varieties are

[1]State Key Laboratory of Crop Gene Resources and Breeding, Institute of Crop Sciences, Chinese Academy of Agricultural Sciences, Beijing 100081, China. [2]Department of Statistics, Purdue University, West Lafayette, IN 47907, USA. [3]Guangdong Key Laboratory of Plant Adaptation and Molecular Design, Innovative Center of Molecular Genetics and Evolution, School of Life Sciences, Guangzhou University, Guangzhou, Guangdong 510006, China. [4]Sanya Nanfan Research Institute of Hainan University, Hainan Yazhou Bay Seed Laboratory, Sanya, Hainan 572025, China. [5]Longping Biotechnology (Hainan) Co., Ltd, Yazhou-Bay Science and Technology City, Sanya, Hainan 572025, China. [6]Heilongjiang Academy of Agricultural Sciences, Harbin, Heilongjiang 150086, China. [7]Jilin Academy of Agricultural Sciences, Changchun, Jilin 130033, China. [8]Tonghua Academy of Agricultural Sciences, Tonghua, Jilin 135007, China. [9]Institute of Cereal and Oil Crops, Hebei Academy of Agriculture and Forestry Sciences, Shijiazhuang, Hebei 050035, China. [10]Liaocheng University, Liaocheng, Shandong 252000, China. [11]Jiangsu Xuhuai Regional Institute of Agricultural Sciences, Xuzhou, Jiangsu 221131, China. [12]Nanjing Agricultural University, Nanjing, Jiangsu 210095, China. [13]Anhui Academy of Agricultural Sciences, Hefei, Anhui 230041, China. [14]Oil Crops Research Institute, Chinese Academy of Agriculture Sciences, Wuhan, Hubei 430062, China. [15]Crops Research Institute, Jiangxi Academy of Agricultural Sciences, Nanchang, Jiangxi 330200, China. [16]Beidahuang KenFeng Seed Co., Ltd, Binxi Economic Development Zone, Harbin, Heilongjiang 150090, China. [17]These authors contributed equally: Chao Qin, Ying-hui Li, Delin Li. ✉e-mail: kongfj@gzhu.edu.cn; qiulijuan@caas.cn; liubin05@caas.cn

restricted to narrow range of latitudes due to their high photoperiod sensitivity, resulting in traits such as maturity time and plant architecture being affected[5,6]. The introduction of new genetic resources has enabled the expansion of cultivation worldwide. In particular, the long juvenile trait (LJ) has been successfully utilized since the 1970s for the breeding of cultivars suitable for lower latitudes (below 22°), making Brazil the largest producer of soybean in the world[7–9].

High latitudes are also essential areas for soybean production. In 2021, China's three northeastern provinces (Heilongjiang, Jilin, and Liaoning at latitudes of over 40°N) accounted for 48.71% of the country's total soybean production (NBSC, 2022). However, the worldwide cultivation of soybean at high latitudes (over 40°N) only produced less than 20% of the total yield, suggesting a great potential to boost global yield by cultivation of soybean in these regions including the far Russia East, Canada, the Northern United States, and the Northeast of China. High latitudes have long-day photoperiods which can induce an extended maturity period and excessive stem growth, impeding the adaptation of soybean varieties to their short frost-free farm seasons. Several flowering time genes, including the early flowering locus (E1-E4)[10–13], Tof5[14], two homologs of PSEUDO-RESPONSE-REGULATOR3, Tof11 and Tof12[15–17], have been reported to be utilized for breeding of soybean varieties suitable for planting in northern regions of China. However, much less is known about the genetic basis of plant height essential for soybean adaptation to high latitudes.

Plant height in soybean is determined by the number of nodes and the length of the internodes, and is highly sensitive to the variation in the light conditions[18]. For instance, long-day photoperiods at high latitude and low blue light (LBL) under high-density planting and intercropping conditions can induce exaggerated stem elongation (ESE) syndrome, decrease the mechanical strength of the stem, result in lodging and increase susceptibility to insects and pathogens[6,19,20]. The above situations seriously constrain the improvement of soybean yield by high-density planting in high-latitude regions. It has been revealed that the blue light receptor GmCRY1s repress the LBL-induced ESE syndrome by stabilizing the bZIP transcription factors STF1 and STF2[21], and modulating the components of the GmCRY1-mediated signaling pathway can improve shade-tolerant traits and improve yield in soybean[22,23].

In this study, we identify a plant height determining gene PH13, which has been selected in breeding soybean cultivars suitable for high latitudes. We show that a Ty1/Copia-like retrotransposon causes a truncation in PH13[H3] protein, resulting in a weaker interaction with GmCOP1a and GmCOP1b, which consequently leads to accumulation of bZIP transcription factors STF1/2 and a short plant stature. We knock out both PH13 and its paralogous gene PHP to generate a ph13php double (phd) mutant that displays an ideal type of shade-resistance. Our findings provide insights into the genetic basis for the breeding of modern soybeans suitable for high latitudes and offer alleles to improve soybean yield through high-density planting.

## Results
### Identification of plant height loci in soybean
To identify genetic loci that can attenuate the extent of ESE, we phenotyped a collection of 2214 previously genotyped soybean accessions[24] for the plant height trait in a normal cultivating system in ten locations over two or three years. The best linear unbiased prediction (BLUP) for each accession was estimated to support association studies (Supplementary Data 1). We conducted a Genome-wide Association Study (GWAS) using 540 spring-sown improved cultivars with 3.47 million SNPs (minor allele frequency >5%) and a Transcriptome-wide Association Study (TWAS) for 488 soybean accessions (Supplementary Fig. 1a) to identify SNPs using FarmCPU[25] and genes associated with plant height. The GWAS identified eleven plant height loci, while the TWAS identified seven genes

(Supplementary Data 2, 3). Of these, fourteen have not been reported previously as soybean height loci and three were refined loci associated with known candidate genes (E2, Dt1 and Dt2)[11,26,27], validating the effectiveness of our strategy to find targeting genes (Fig. 1a, b). Only one gene (Glyma.13G276700) located on chromosome 13 was identified by both GWAS and TWAS. Hence, we named this candidate gene as Plant Height 13 (PH13) for further analysis. The surrounding genetic region of PH13 has repeatedly been identified as a plant height-associated QTL in previous studies[28–32], supporting a role of PH13 in regulating plant height.

The leading single nucleotide polymorphism (SNP), Gm13-37757704, identified by GWAS has a physical distance of 55,493 bp from PH13 and a linkage disequilibrium R$^2$ of 0.86 with the exonic SNP (Gm13-37816013) within PH13 (Fig. 1a and Supplementary Fig. 1b). The TWAS identified the sixth exon of PH13 within this locus (Fig. 1b). Since FarmCPU considers associated markers as covariants, which can lead to false negatives of neighboring coexpressed genes of PH13 in TWAS, we also used a mixed linear model to perform TWAS, which confirmed that PH13 is the only gene within this locus that displayed an association signal (Supplementary Fig. 1c).

### A homologue of SUPPRESSOR OF PHYA is the candidate of PH13 gene
The PH13 gene encoding a WD40 protein which is likely homologous to the suppressor of the phyA-105 (SPA) family protein in Arabidopsis[33]. Phylogenetic analysis indicated that PH13 is grouped closely with SPA3/4 protein (Supplementary Fig. 2a and Supplementary Data 4). We then transformed the 35S::PH13-3×Flag construct into the spa134 mutant[34]. The phenotypic results indicated that ectopic expression of PH13 can at least partially rescued the dwarf phenotype of the spa134 mutant at both seedling and adult vegetative stage, supporting that PH13 is homologous to the SPA family proteins (Supplementary Fig. 2b–e).

The TWAS analysis of 488 accessions revealed two forms of PH13 transcripts (Supplementary Fig. 3), one of which had no detectable expression from shortly after the beginning of the fifth exon to the 3' end (Fig. 1c). To understand the cause of this difference, we examined the coding DNA sequence (CDS) of PH13 in 1254 accessions by re-sequencing and PCR assay (Supplementary Fig. 4a, b). Besides the exonic SNP (Gm13-37816013), a 5404 bp fragment containing two 431 bp long terminal repeats (LTRs) and a 3984 bp open reading frame (ORF) encoding Gag-protease-integrase-RT-RNaseH domains was found at the start of the fifth exon. This fragment belongs to the Ty1/Copia-like retrotransposon (Fig. 1d, Supplementary Fig. 5 and Supplementary Data 5, 6).

Three main haplotypes (PH13[H1]-PH13[H3]) were detected in the PH13 CDS of the 1254 accessions based on the SNPs and insertion fragment variation (Table 1). The haplotype 3 (PH13[H3]) harbors the Ty1/Copia-like retrotransposon insertion, which was predicted to produce a truncated PH13 protein (742 amino acids) lacking the 3' part of the WD40 domain (Supplementary Fig. 6). This hypothesis was confirmed by immunoblot results showing that the molecular weight of PH13[H3]–3×Flag was about 10 kDa lower than that of PH13[H1]–3×Flag or PH13[H2] –3×Flag when ectopically expressed in tobacco leaves (Supplementary Fig. 7). Moreover, PH13[H3] accessions were found to have significantly reduced plant height compared to accessions carrying PH13[H1] and PH13[H2] under field conditions in ten field locations over two or three years (Fig. 1e and Supplementary Data 7), suggesting that the retrotransposon insertion in PH13[H3] is responsible for the reduction in plant height.

### Genetic validation of PH13 function in regulating plant height
Tissue-specific expression analysis showed that PH13 was expressed at a higher level in the shoots and stems compared to other tested tissues, including roots, cotyledons, unifoliate and trifoliate leaves, and

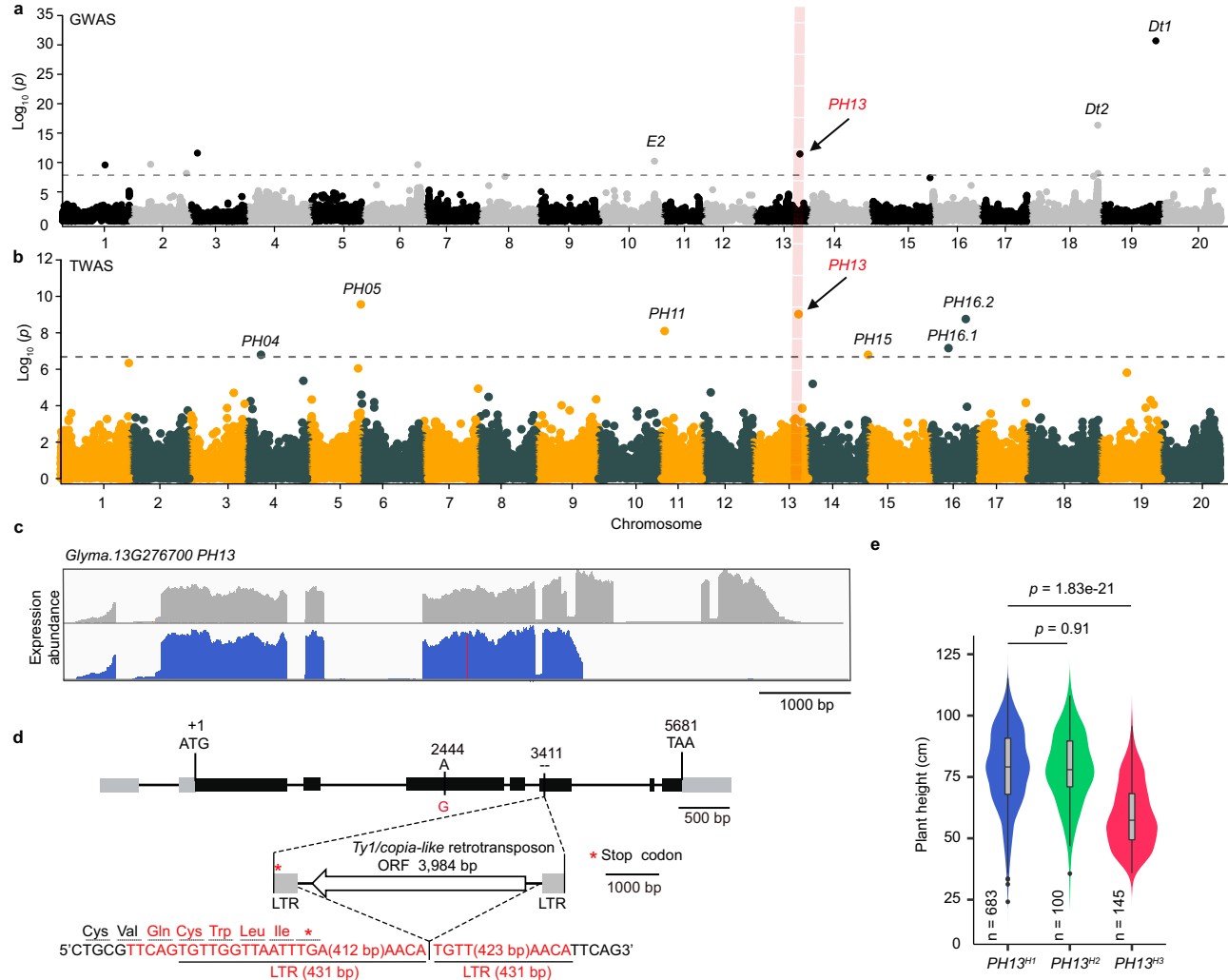

**Fig. 1 | Identification of *PH13* as a major QTL for plant height in soybean.**
**a**, **b** Manhattan plots for GWAS (540 accessions) and TWAS (488 accessions) of soybean plant height (Supplementary Data 1). Each dot represents one SNP in GWAS (**a**) or gene/exon in TWAS (**b**). The p values derived from association analyses conducted using FarmCPU were log-transformed, with the gray dashed line representing the Bonferroni correction threshold for multiple test adjustments. Previously identified genes are labeled in black, and *PH13* is labeled in red. The green and orange dots are arranged in alternation to distinguish them from different chromosomes. **c** Two types of transcripts associated with plant height variation were shown. The transcripts with normal expression levels are displayed at the top, while those with truncated expression due to the insertion of a *Ty1/Copia*-like retrotransposon are shown below. Five samples from each category were randomly selected and pooled for alignment visualization. **d** Schematic representation of the *PH13* candidate gene and the insertion site of a *Ty1/Copia*-like retrotransposon. Solid and shaded boxes in the gene structure represent exons and UTRs, respectively. The retrotransposon diagram is shown at the bottom, with LTR representing long terminal repeats and ORF representing open reading frame. The red "G" (base 2444) represents the nonsynonymous mutation derived from exonic SNP. The red letters below denote flanking nucleotides and amino acids derived from the retrotransposon insertion. **e** Distribution of plant height BLUP (Best Linear Unbiased Prediction) for each haplotype. The BLUP values were calculated using the plant height data of natural populations in ten field environments over two or three years (Methods-GWAS and TWAS assays). The box plot shows the 25th to 75th percentile range, with a black line indicating the median. The whiskers extend to cover a range of 1.5 times the interquartile range, and black dots represent outliers. p values obtained from unpaired, two-tailed Student's t-tests (Supplementary Data 7).

flowers, supporting *PH13* is involved in regulation of plant height (Supplementary Fig. 8). To investigate its biological role in soybean, two independent gRNAs targeting the first exon were used to generate loss-of-function mutants in the Williams 82 background (W82^H1, harboring *PH13^H1*) using CRISPR/Cas9 technology. Three independent mutants, *ph13-1*, *ph13-2*, and *ph13-3*, were identified with base deletion or/and insertion (Supplementary Fig. 9). The phenotypic results demonstrated that the loss-of-function of *PH13* caused a significant dwarf phenotype with a more than 30% reduction in plant height compared to the wild type (WT) W82^H1 (Fig. 2a, b and Supplementary Fig. 10a, b). Additionally, we generated three independent *PH13^H1*−3×*Flag* overexpression lines (*H1-OE1*/TL1^H3, *H1-OE2*/TL1^H3 and *H1-OE3*/TL1^H3) in the TL1^H3 background, which exhibited a significant

increase in plant height (Fig. 2a and Supplementary Fig. 10d–f). The reduction in plant height of the *ph13* mutants was mainly due to a decrease in internode length (Fig. 2b and Supplementary Fig. 11) which was correlated with the reduced cell length (Supplementary Fig. 12). Moreover, the *ph13* mutants showed 5 days earlier flowering time and 2 fewer nodes than W82^H1, which also contributed to the reduction in plant height (Supplementary Figs. 10c and 11b).

To assess the functionality of *PH13^H3*, three independent mutants were generated using CRISPR-Cas9 technology in a Tianlong 1 back-grounds (TL1^H3, harboring *PH13^H3* with the retrotransposon insertion). Phenotypic analysis revealed that the mutants had a slightly reduced plant height of around 10% compared to the WT TL1^H3 (Fig. 2a, b), indicating that the *PH13^H3* retained some of its ability to promote plant

height. In addition, near-isogenic lines (NILs) were created by crossing the W82$^{H1}$ cultivar (with an average plant height of 110 cm) with the TL1$^{H3}$ cultivar (with an average plant height of 80 cm) (Supplementary Fig. 13). The resulting NIL$^{H3}$ plants had a significantly lower plant height than that of NIL$^{H1}$ (Fig. 2c). Collectively, these results demonstrate that *PH13* functions as a plant height enhancer by promoting stem elongation and increasing node number, and the *Ty1/Copia*-like retrotransposon insertion had partially comprised the function of *PH13*$^{H3}$ in regulating plant height.

### *PH13*$^{H3}$ was artificially selected for the improvement of soybean at high latitude

Given that plant height is one of the essential agronomic traits determining yield[35-37], we investigated whether *PH13* alleles have been utilized during soybean domestication and improvement. We analyzed the frequency and geographic distribution of the three main haplotypes among 121 accessions of *G. soja*, 715 landraces, and 418 improved cultivars, all with known genotypes and origin sites around the world

### Table 1 | Three major haplotypes of *PH13* identified in 1254 soybean accessions

| Haplotypes | Position in *PH13* | |
|---|---|---|
| | **2444** | **3411** |
| *PH13*$^{H1}$ | A | – |
| *PH13*$^{H2}$ | G | – |
| *PH13*$^{H3}$ | G | 5404 bp insertion |

The haplotype analysis of *PH13* was conducted using a core collection of 1254 soybean accessions (Supplementary Data 7). The haplotype 3 of *PH13* (*PH13*$^{H3}$) harbors a 5404 bp retrotransposon insertion.

(Supplementary Data 7). We found that *G. soja* predominantly contain *PH13*$^{H1}$ (99.2%) with a small amount of *PH13*$^{H2}$ (Fig. 3a). Meanwhile, *PH13*$^{H3}$ was absent in *G. soja* but occurred at a low frequency of 1.4% in landraces (Fig. 3a). Interestingly, the proportion of *PH13*$^{H3}$ accessions among all improved cultivars increased to 32.7%, suggesting that the *PH13*$^{H3}$ allele was subject to intensive artificial selection during genetic improvement.

The increase in the proportion of *PH13*$^{H3}$ in improved cultivars compared to landraces prompted us to investigate its geographical distribution in the world. We found that a large proportion of the *PH13*$^{H3}$ accessions were concentrated in high-latitude regions, while the other two haplotypes were dispersed more evenly (Fig. 3b). This geographical bias was further analyzed in China where the proportion of *PH13*$^{H3}$ among all cultivars increased from near zero in the lower latitude regions (below 40°N) to 48.7% in the higher latitude regions (above 40°N) (Supplementary Fig. 14a–d). Out of 78 cultivars carrying *PH13*$^{H3}$, 76 were located in higher latitude regions (Supplementary Fig. 14e), demonstrating that the *PH13*$^{H3}$ allele has been used successfully in breeding programs to improve soybean adaptation in high latitudes.

### The protein products of *PH13*$^{H3}$ have reduced binding affinity with two GmCOP1 E3 ubiquitin ligases

PH13, being a SPA homologous protein in soybean, possesses a conserved kinase domain, coiled-coil domain, and WD40 domain, while the protein products of *PH13*$^{H3}$ lacks the intact WD40 domain (Supplementary Figs. 6 and 7). In order to assess the impact of the retrotransposon insertion on PH13's function, we conduced RT-qPCR analysis on W82$^{H1}$ and TL1$^{H3}$ cultivars. The results revealed that the insertion did not affect the transcription of the sequence before the insertion site, but abolished the transcription of the carboxyl terminus

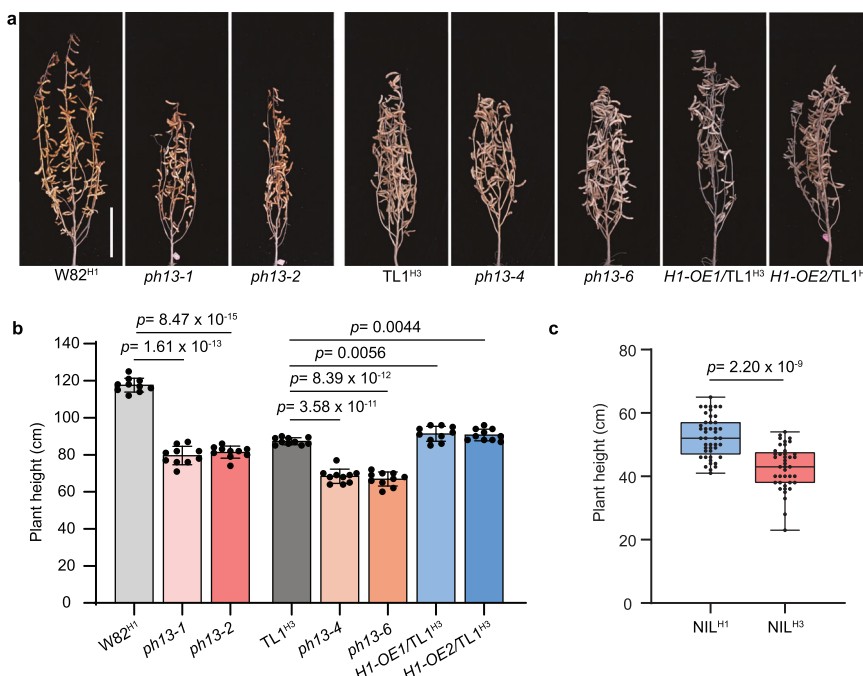

**Fig. 2 | Genetic confirmation of *PH13* as a plant height regulator. a** Gross photos of the indicated lines grown under natural field conditions in the summer of Beijing. The CRISPR/Cas9-engineered mutants *ph13-1* and *ph13-2* are in the W82$^{H1}$ (carrying *PH13*$^{H1}$) background, and *ph13-4* and *ph13-6* are in the TL1$^{H3}$ (carrying *PH13*$^{H3}$) background. The *PH13*$^{H1}$ overexpression lines (*H1-OE1*/TL1$^{H3}$ and *H1-OE2*/TL1$^{H3}$) are in the TL1$^{H3}$ background. Scale bar, 20 cm. **b** Plant height of the indicated lines as shown in **a**. Data are mean ± SD (*n* = 10 biologically independent plants). **c** Plant height of the near-isogenic lines (NILs) carrying homozygous H1 (NIL$^{H1}$) and homozygous H3 (NIL$^{H3}$) under natural field conditions in Beijing. The NILs were derived from the hybrid combination between W82$^{H1}$ and TL1$^{H3}$ (Supplementary Fig. 13). The two ends of the box plot and the upper, middle, and lower box lines represent the upper edge, lower edge, median, and two quartiles of values. Data are mean ± SD (*n* = 45 and 41 biologically independent plants for NIL$^{H1}$ and NIL$^{H3}$, respectively). Above *p* values were calculated by unpaired, two-tailed Student's *t*-tests. Source data are provided as a Source Data file.

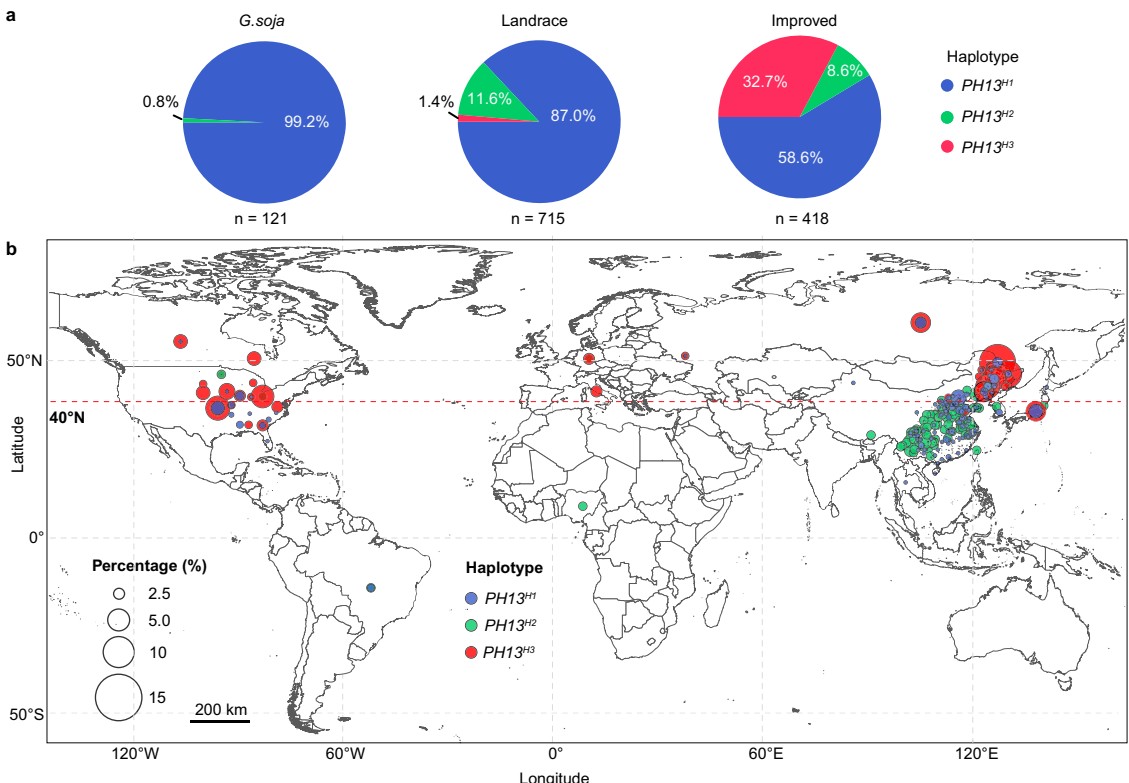

**Fig. 3 | The evolution and geographical distribution of *PH13* different haplo-types. a** The pie charts represent the percentage of accessions with different haplotypes in wild soybean, landrace, and improved cultivars. **b** The global geographical distribution of 1133 soybean accessions (including landrace and improved cultivars) carrying *PH13^H1^*, *PH13^H2^*, and *PH13^H3^* (Supplementary Data 7) is shown using circle. The color of the circle represents the type of germplasm, and the size of the circle represents the percentage of germplasm in respective location.

(Fig. 4a, b), which is consistent with the expression patterns observed in natural population (Fig. 1c). Further analysis of the subcellular location of different PH13 haplotypes showed that the retrotransposon did not alter the location of the truncated PH13^H3 protein in both nucleus and cytoplasm (Supplementary Fig. 15).

The SPA family is known to suppress photomorphogenesis, as part of a complex with COP1, serving as E3 ubiquitin ligases to target multiple transcription factors for degradation in *Arabidopsis*[38–40]. A previous study showed that two soybean COP1 orthologs, GmCOP1a and GmCOP1b, play a pivotal role in controlling plant height in soybean[22]. Here we found that the diurnal transcription pattern of *PH13* is in line with that of *GmCOP1a* and *GmCOP1b* in soybean, peaking at dawn and declining at dusk (Fig. 4c), suggesting that PH13 may act as an evolutionarily conserved factor to form a complex with these GmCOP1s to degrade target proteins in soybean. This prompted us to investigate whether the retrotransposon affects the interaction of PH13 with GmCOP1s. Yeast-two hybrid (Y2H) and Co-Immunoprecipitation (Co-IP) experiments revealed that these GmCOP1s strongly interact with PH13^H1 and with PH13^H2 but weakly with PH13^H3 (Fig. 4d–f and Supplementary Fig. 16). Domain-specific Y2H assays indicated that PH13 interacts with GmCOP1b via their coil-coil domains, but the WD40 domain of PH13 can significantly enhance their interaction strength (Supplementary Fig. 17a–c). These results together demonstrate that the absence of the WD40 domain in PH13^H3 reduces the interaction strength between PH13 and GmCOP1s in soybean.

## PH13 and its paralogue PHP function together to decrease STF1/2 abundance

Given previous evidence that GmCOP1s mediate the degradation of STF1/2 transcription factors which are homologous to *Arabidopsis* HY5 and responsible for inhibiting stem elongation in legume[21,22,41,42], we sought to determine whether the altered interaction of the truncated PH13 with GmCOP1s affects the accumulation of STF1/2 proteins. Our results showed that the overall protein levels of STF1/2 increased by 39–144% during a diurnal cycle in the NIL^H3 line compared to NIL^H1 line grown under long day condition in growth chamber (Fig. 4g, h and Supplementary Fig. 18). This indicates that the insertion retrotransposon in NIL^H3 enhances the accumulation of STF1/2. Additionally, the pattern of STF1/2 protein accumulation, which peaked in the middle of the day (Fig. 4h), was opposite to that of the *GmCOP1s* and *PH13* transcripts peaking at night (Fig. 4c). This suggests that the GmCOP1s-PH13 E3 ligases play a role in modulating the abundance of the STF1/2 protein in response to photoperiod.

While the abundance of the STF1/2 protein increased in the NIL^H3 line, its expression pattern remained unchanged and its level still decreased during the night period (Fig. 4h and Supplementary Fig. 18), suggesting a possibility that some homologous genes may act redundantly or additively with *PH13* to regulate the degradation of STF1/2. Indeed, the soybean genome contains a *PH13* paralogue *Glyma.12G224600*, which we hereafter name *PHP*, encoding a protein with 96.6% amino acid similarity to PH13 (Supplementary Figs. 2a and 19). We then generated multiple *php* and *ph13/php* double (*phd*) mutant lines using CRISPR/Cas9 technology in the TL1^H3 background (Supplementary Fig. 20). Phenotypic analysis revealed a progressive reduction in plant height in the order of TL1^H3, *ph13*, *php*, and *phd* (Supplementary Fig. 21), indicating that the *PH13* and *PHP* genes act cooperatively in regulating plant height. We observed at least 3-fold increase in STF1/2 protein abundance in the *phd-1* mutant compared to the WT TL1^H3 (Supplementary Fig. 22) especially at night. To verify whether *PH13* and *PHP* act as upstream regulators of *STF1/2*, we crossed the *phd-1* mutant with the *stf1/2* mutant[43] (Supplementary Fig. 23a).

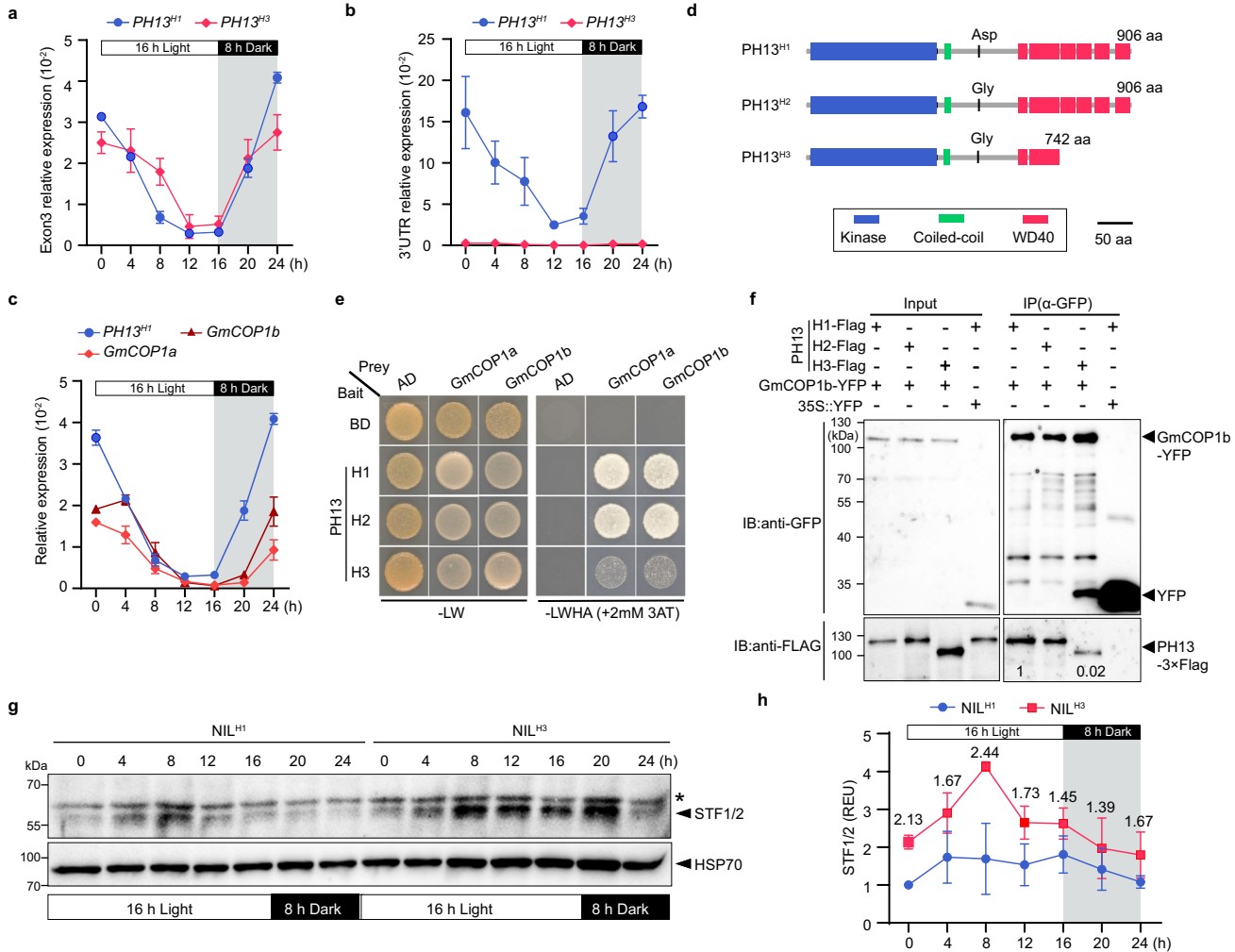

**Fig. 4 | Comparison of transcripts and protein activities of different haplotype of _PH13_.** Transcript levels of _PH13^H1_ in W82 and _PH13^H3_ in TL1 respectively, were measured using primers detecting the expression of exon 3 (**a**) and 3'UTR (**b**). **c** Transcript levels of _PH13^H1_, _GmCOP1a_, and _GmCOP1b_ in W82. The second trifoliate leaves of 20-day-old seedlings grown under long-day conditions were collected at 4-hour intervals for RT-qPCR analysis. Data are mean ± SD (_n_ = 3 biologically independent replicates) calculated relative to _GmActin_. **d** Protein structure of each haplotype. aa, amino acids. **e** Auxotrophic assays showing the interactions between different PH13 haplotypes with GmCOP1s. Yeast cells transformed with indicated constructs were selected on -LW (lacking Leu and Trp) or -LWHA (lacking Leu, Trp, His and Ade) medium. AD, GAL4 activation domain; BD, GAL4 DNA-binding domain. **f** Co-Immunoprecipitation (Co-IP) assay showing the interaction between each PH13 haplotype with GmCOP1b in tobacco leaves. The indicated constructs were co-transformed into tobacco which were then incubated at 25 °C in dark for 12 h and grown under white light (WL, 80 μmol m⁻² s⁻¹) for 36 h. The immunoprecipitates were detected using anti-GFP (at a 1:2500 dilution) and anti-Flag (at a

1:2500 dilution) antibodies, respectively. Empty vector (_35 S::YFP_) was used as a negative control. Numbers at bottom represent the relative IP efficiency, calculated as (IP-PH13/Input-PH13)/(IP-GmCOP1b/Input-GmCOP1). A representative result of three independent replicates is shown. **g** Immunoblots showing STF1/2 protein levels in the NIL^H1 and NIL^H3 lines under diurnal conditions. The first trifoliate leaves of 15-day-old seedlings grown under long-day condition were collected at 4-hour intervals. The membrane was probed with the anti-STF1/2 antibody (at a 1:1000 dilution), stripped, and then probed with the anti-HSP70 antibody (at a 1:10,000 dilution). The asterisk indicates a non-specific band. **h** The relative expression level of STF1/2 proteins represented by REU (Relative Expression Unit) was calculated by the formula [STF1/2] / [HSP70], in which 'STF1/2' and 'HSP70' indicate the digitized band intensity of STF1/2 or HSP70 in each sample collected at respective time point. The REU of STF1/2 in NIL^H1 at ZT0 was arbitrarily set to 1. Data are shown as means ± SD of three biological replicates (Supplementary Fig. 18). Source data are provided as a Source Data file.

Phenotypic assessment showed partial rescue of the dwarf phenotype of the _phd-1_ mutant by the _stf1/2_ mutant (Supplementary Fig. 23b, c), suggesting that STF1/2 act as downstream targets of _PH13_ and _PHP_ to influence plant height.

### _PH13_ and _PHP_ redundantly regulate LBL-induced exaggerated stem elongation

Considering that low blue light (LBL) triggers degradation of STF1/2 and leads to ESE syndrome in soybean[21], we hypothesize that the increased abundances of STF1/2 in the _ph13_, _php_ and _phd_ mutants might result in an appropriate extent of stem elongation for high yield under shade conditions. To test this, we compared the performance of

each mutant line with the WT TL1^H3 in response to simulated shade regimes (Fig. 5a and Supplementary Fig. 24). Our results showed that LBL efficiently induced ESE syndrome in the _ph13_ and _php_ mutants as well as in TL1^H3, but not in the _phd_ mutant (Fig. 5b).

The blue light receptor, GmCRY1s, was observed to mainly mediated LBL-induced shade avoidance syndrome (SAS) in soybean[21]. Next, we crossed the CRISPR-Cas9-engineered _Gmcry1s_ quadruple (_Gmcry1s-qm_) mutant[21] displaying constitutive ESE syndrome, with the _phd-1_ mutant carrying stocky phenotype, to obtain the _phd-1/Gmcry1s-qm_ sextuple mutant (Supplementary Fig. 25a). We found that the _phd_ mutations partially rescued the ESE syndrome of the _Gmcry1-qm_ mutant in field conditions (Supplementary Fig. 25b–d), suggesting that

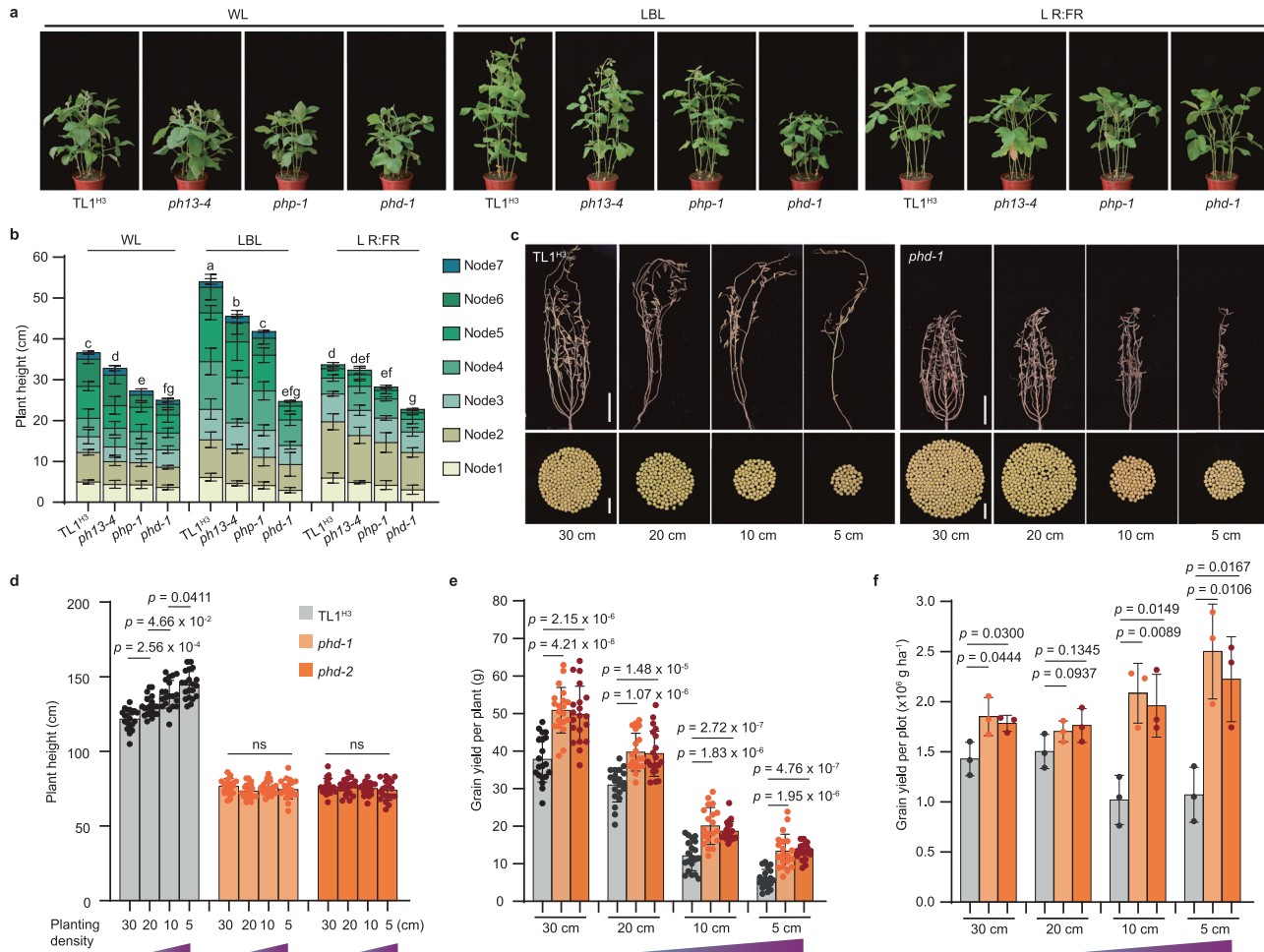

**Fig. 5 | CRISPR/Cas9 targeting of the *PH13* gene and its paralog (*PHP*) abolished the LBL induced ESE and enhanced yield under different planting densities.** **a** Representative images of the *php-1*, *ph13-4*, and *phd-1* mutants and WT TL1 under different light regimes. Seedlings were grown under white light (WL), LBL, and WL plus far-red (L R:FR) for 15 days after de-etiolation with white light under long-day conditions. The intensity of photosynthetically active radiation (PAR) was maintained at approximately 500 μmol m⁻² s⁻¹ (Supplementary Fig. 24). Scale bar corresponds to 20 cm. **b** Plant height and internode length of the indicated lines shown in **a**. Data are means ± SD (*n* = 6 biologically independent plants for **b**). Different letters indicate statistically significant differences as determined by two-way ANOVA with two-sided Tukey test at the 0.05 level. **c** Gross photos of the WT TL1 and *phd-1* mutant plants (upper panel, Scale bar, 20 cm) and the whole seeds (bottom panel, Scale bar, 2 cm) produced by respective plant grown with 30 cm, 20 cm, 10 cm, or 5 cm plant space, respectively. Statistical analysis of the plant height (**d**), grain yield per plant (**e**), and grain yield per plot (**f**) of each indicated line. Data are means ± SD (*n* = 20 biologically independent plants for **d** and **e**; *n* = 3 biologically independent field plots for **f**). *p* values are from unpaired, two-tailed Student's *t*-tests. Source data are provided as a Source Data file.

*PH13* and *PHP* may be genetically downstream of GmCRY1s in regulating of LBL-induced stem elongation. Furthermore, the *phd* mutant was found to display a semi-dwarf phenotype due to decreased internode length, rather than reduced node number in comparison with TL1^H3, under LBL conditions (Fig. 5b). Moreover, the *phd* mutant was distinguished by its thicker stem compared to other lines under all light conditions (Fig. 5a and Supplementary Fig. 26). Collectively, these results demonstrate that the *phd* mutant is characterized by ideal shade-resistant traits, which may be advantageous for high-density planting or intercropping at high latitudes.

## Knockout of *PH13* and *PHP* enable TL1^H3 to adapt to high latitudes

Generally, the allopatric planting of lower-latitude soybean cultivars in higher-latitude region can induce excessive stem growth and lead to severe lodging due to the extended vegetative phase (Supplementary Fig. 27). To verify if the *phd* mutants could be utilized to enhance the adaptability of cultivars to higher latitudes, we conducted field trails in Changchun (125°19′E, 43°53′N) and Xuchang (104°31′E, 34°10′N) and compared the extent of stem elongation between the *phd* mutants and

TL1^H3 (Supplementary Fig. 28a, b). The results showed that the *phd* mutant had a lower fold change in stem elongation between the two latitudes than TL1^H3 (Supplementary Fig. 28c). Moreover, the *phd* mutants had a lower lodging rate, earlier maturity time, more branches, and a higher pod number per plant than the *ph13*, *php* mutants, and WT TL1^H3 (Supplementary Fig. 29). In addition, we observed that the *phd* mutants exhibit a larger root system at seedling stage in growth chamber and at maturity stage in Beijing field conditions (Supplementary Fig. 30). Above results further suggesting the potential of using the *phd* mutant to improve the adaptability of soybean to high-latitude regions.

## The *phd* mutants is suitable for high-density planting or intercropping

We evaluated the yield performance of the *phd* mutants at different planting densities (30 cm, 20 cm, 10 cm, and 5 cm plant space, equivalent to approximately 67,000, 100,000, 200,000, and 400,000 plants per hectare) in Changchun (Supplementary Fig. 31). The results showed that the main stem length of TL1^H3 increased with the increase of planting density (IPD), resulting in severe lodging at all planting

densities, whereas the *phd* mutants were insensitive to IPD and no lodging occurred (Fig. 5c, d and Supplementary Figs. 31c, 32c). While the yield per plant decreased with the IPD for TL1$^{H3}$, the *phd* mutants consistently outperformed it (Fig. 5e and Supplementary Fig. 32). The plot yields of the *phd* mutants increased significantly in response to the IPD, while remaining unchanged or even decreased for TL1$^{H3}$ (Fig. 5f). Moreover, the *phd* mutant lines showed better performance than a local elite cultivar Jiyu202 (JY202) at high planting density (400,000 plants per hectare) (Supplementary Fig. 33a). Despite being shorter than JY202, the *phd* mutants did not display any difference in node number (Supplementary Fig. 33d). Additionally, the *phd* mutants had a thicker stem, lower lodging rate, more branches, more pods per plant, as well as a 15.8% increase in grain yield per plant compared to JY202 (Supplementary Fig. 33b–h).

Furthermore, we evaluated the shade-tolerant ability of the *phd* mutants under a maize-soybean relay intercropping system at even higher latitudes in Harbin (127°50′E, 45°70′N). The results showed that the *phd* mutants were significantly shorter than TL1$^{H3}$ as well as LK317 and LK18-842 (Supplementary Fig. 34a–c), two elite cultivars used for maize-soybean intercropping in Harbin. Moreover, lodging did not occur in the *phd* mutants, while nearly 100%, 40% and 80% lodging rates were observed in TL1, LK317 and LK18-842, respectively (Supplementary Fig. 34d). The plot yield of the *phd* mutants was at par with that of LK317 and LK18-842 (Supplementary Fig. 34e). These results together demonstrate that knockout of *PH13* and *PHP* can improve the adaptability of TL1$^{H3}$, a cultivar of mid latitude, to higher latitude for high density planting or intercropping.

## Discussion

The global demand for soybeans is expected to continue rise in the future due to several factors such as the rising consumption of meat and soy-based health products, growing populations, and a more favorable policy of biofuels. Currently, the majority of soybean cultivation is located in mid to low latitude countries, including the United States, Brazil, and Argentina, which together contribute 80% of the total global yield[44]. Thus, there present a great opportunity for soybean production in high-latitude regions, such as Canada, northern US, northern China, and Russia, which could potentially serve as the primary source of increased soybean production in the future.

However, high-latitude regions have short frost-free periods and a long-day photoperiod, which can lead to excessive stem elongation and lodging. Those require the development of soybean varieties with a shorter growth period and reduced plant height. In this study, a natural variant of the *PH13* gene (*PH13$^{H3}$*) was identified that has been unintentionally utilized in the breeding of modern soybean varieties in northern latitude regions. The underlying mechanism of the reduced plant height was determined to be the insertion of the *Ty1/Copia*-like retrotransposon in the *PH13* gene, causing a partial loss of the WD40 domain in the PH13 protein (Fig. 4d) and weakening its interaction with GmCOP1s and allowing the accumulation of downstream transcription factors STF1/2 (Fig. 4g, h), which subsequently inhibit internode elongation. The interaction between PH13 and GmCOP1b is mediated by their coil-coil domains, and enhanced by the WD40 domain of PH13. This is likely different from *Arabidopsis*, where the interaction between SPA1 (a homologous protein of SPA3/4 and PH13) and COP1 is also mediated by their coil-coil domains, and the absence of the WD40 domain of SPA1 does not impair the interaction with COP1[45]. Furthermore, COP1 and SPAs have been reported to co-localize and function together within the nucleus in *Arabidopsis*[46,47], whereas PH13 lacks an NLS (Nuclear Localization Signal) and is uniformly distributed in both the nucleus and cytoplasm (Supplementary Figs. 15 and 19). This suggests that PH13 may be involved in a distinct mechanism for regulating soybean growth and development.

Although the *PH13$^{H3}$* allele has been strongly selected in northern latitudes, its ability to reduce plant height may be insufficient and needs to be combined with early flowering loci to suit the environmental requirements of higher latitude regions[14,48,49]. This is evidenced by the TL1$^{H3}$ cultivar, which harbors the *PH13$^{H3}$* haplotype but exhibits severe ESE and lodging in the northern regions (Fig. 5c and Supplementary Fig. 27). To address this problem, the *PH13* gene and its paralog gene *PHP* were simultaneously eliminated to produce the *phd* mutants in TL1$^{H3}$ background, which exhibit both reduced plant height and early maturity suitable for planting at high latitudes (Fig. 5d and Supplementary Fig. 29f). Additionally, many current soybean variates in the north latitudes are not suitable for high-density planting and soybean-maize intercropping which can exacerbate the problem of lodging caused by long-day photoperiods[20,50]. Low blue light (LBL) is the main shade avoidance signal that induces the ESE syndrome in soybean[21]. The *phd* mutant is insensitive to LBL and lodging resistant (Fig. 5a, b), thus significantly increasing yield under high-density planting conditions at high latitudes (Fig. 5e, f). Moreover, the *phd* mutations can also benefit intercropping by reducing the lodging-induced yield reduction (Supplementary Fig. 34d, e). Although the *phd* mutant in the TL1 background exhibited lower yield potential compared to the elite cultivars LK317 and LK18-842 in Northern China under norm planting conditions, it displayed a lower lodging rate and a similar grain yield potential relative to the elite cultivars LK317 and LK18-842 under maize-soybean relay intercropping conditions. In conclusion, this study identified a pair of plant height regulatory genes that can improve soybean adaptation to high latitude and provided a strategy for breeding high-yield varieties suitable for dense planting and intercropping at high latitudes (Fig. 6).

## Methods
### Plant materials and growth conditions
The high-quality genome sequences of 2214 accessions have been published before[24]. Among these materials, 540 improved cultivars with phenotype data were utilized for GWAS, resulting in the discovery of an association between genotype and phenotype variation. Furthermore, TWAS analysis uncovered an association between gene between gene expression and phenotype variation in 488 soybean accessions that possessed both phenotype and expression data. Detailed information of these materials is available in Supplementary Data 1.

In this study, the soybean (*Glycine max* (L.) Merr.) cultivar Tianlong 1 (TL1) and Williams 82 (W82) served as control groups to generate transgenic lines. To evaluate plant height, we cultivated WT, *ph13*, *php*, and *phd* mutants, as well as *PH13* overexpression lines under long day conditions (16 h light/8 h dark at 26 °C) in a controlled growth chamber. For the field experiment, the aforementioned transgenic materials were grown naturally from May to October in three different locations: the Institute of Crop Science, Chinese Academy of Agricultural Science, Beijing (116°23′E, 39°54′N), Jilin Agricultural University, Changchun (125°19′E, 43°53′N), Beidahuang KenFeng Seed Co., Ltd, Harbin (127°50′E, 45°70′N).

To assess the plant density effects, four treatments with three replicates each were conducted in Changchun. Seeds were sown in May with varying plant spacings (30 cm, 20 cm, 10 cm, and 5 cm, resulting in plant densities of 66,700, 100,000, 200,000, and 400,000 plants per hectare, respectively). The plots were organized in 3 m long rows with 0.5 m between each row, covering a total area of 7.5 square meters. Harvesting was performed in October 2021.

For the maize-soybean relay intercropping experiments, seeds were sown in early May in Harbin, with a row-to-row distance of 40 cm between maize plants, and between soybean plants, and 60 cm between maize and soybean. The soybean was spaced 5 cm apart in rows of 5 m in length (equivalent to 400,000 plants per hectare), while the maize was spaced 10 cm apart in rows of 5 m in

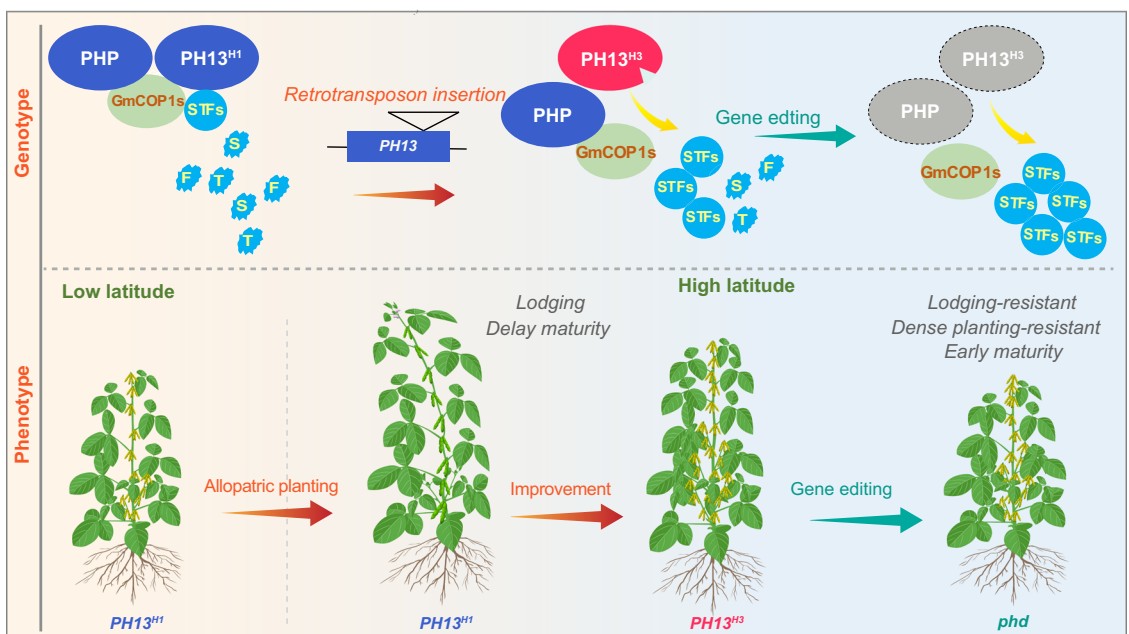

**Fig. 6 | A proposed working model of *PH13* and *PHP* for breeding of elite cultivar suitable for high density planting in high latitudes.** The insertion of the *Ty1/Copia*-like retrotransposon in the *PH13 H3* gene attenuates the interaction between PH13 and GmCOP1s and increases the abundance of STFs protein, resulting in a stocky architecture in the natural population. The soybean accessions harboring *PH13 H3* were selected during improvement and utilized by modern breeders at high latitude. The *phd* double mutants (harbors mutation in both *PH13* and *PHP*) display excellent shade traits under high-density planting conditions, which can be further utilized to improve the yield at high latitudes.

length. To maintain uniformity, all lines under different densities were manually planted with 3 seeds per hole. Once the seeds geminated and the unifoliate leaf fully unfolded, only one healthy seedling was retained per hole. Failed seedlings were removed and replaced with a healthy seedling of same genotype via transplanting to maintain uniformity. The plot size was 19.7 square meters, and two biological replicates were conducted for each experimental material. Harvesting was performed in October 2022. At the R8 stage (when 95% of pods had reached maturity), the following agronomic traits were measured: plant height, length of the internodes, number of branches, node number, height of the center of gravity point, pods per plant, lodging rate, grain yield per plant, and grain yield per plot were measured. Soybean lodging resistance was evaluated based on the height of the center of gravity point, a measure of the balance point at which mature plants were placed in a horizontal position. At least ten randomly selected plants within each plot were included in the phenotypic analysis for each measured trait.

## GWAS and TWAS assays

A panel of 857 soybean accessions (Supplementary Fig. 1a) were genotyped and phenotyped for plant height for association analysis in a previous study[24]. The plant materials were planted in ten different environments and evaluated for a minimun of two years to record plant height data. These environments included Harbin (45°45′N, 126°41′E) from 2017 to 2019, Changchun (43°88′N, 125°25′E) in 2018 and 2019, Tonghua (41°74′N, 125°94′E) from 2017 to 2019, Shijiazhuang (38°05′N, 114°52′E) in 2018 and 2019, Liaocheng (36°46′N, 115°99′E) from 2017 to 2019, Xuzhou (34°27′N, 117°19′E) in 2018 and 2019, Nanjing (31°05′N, 118°78′E) in 2018 and 2019, Hefei (31°88′N, 117°17′E) in 2018 and 2019, Wuhan (30°58′N, 114°32′E) in 2018 and 2019, and Nanchang (28°32′N, 116°1′E) from 2017 to 2019. Each cultivar was planted in a 1.8 m × 0.8 m plot with two rows and a spacing of 10 cm between seedlings in each row. The soybean lines' plant height Best Linear Unbiased Predictions (BLUPs) were calculated using a mixed linear model (**1**)[51]. Here, Y represents the observed plant height, X is the fixed effects, β is the vector of fixed effect coefficients, Z is the random

effects, u is the random effect coefficients, and e is the residual errors.

$$Y = X\beta + Zu + e \tag{1}$$

In the model, the fixed effect represents the mean plant height across all lines, locations, and years. Random effects account for variations within each line, location, year, and their interactions (Line: Location and Line: Year). To address incomplete data in some lines across experimental trials, both Year and Location were treated as random effects.

The re-sequencing of 2214 soybean accessions generated 8,785,134 SNPs, which were imputed by beagle[24]. After filtering the imputed SNPs for a minor allele frequency of >5% among the genotyped and phenotyped 540 soybean accessions, the remaining 3,469,934 SNPs were retained and used for plant height GWAS. The GWAS was performed using the fixed and random model Circulating Probability Unification (FarmCPU)[25], with population structure controlled by the first three components from SNPs principal components analysis (PLINKv1.90)[52]. The resulted *p*-values were adjusted with Bonferroni correction at a level of α = 0.05, resulting in a cutoff of 1.44E−08.

A natural panel (PRJCA014188) of previously published RNA-Seq data from tissues above cotyledonary node at the V2 stage was employed for TWAS. A total of 488 soybean accessions, possessing both expression and phenotype data, were included in the analysis and underwent TWAS using FarmCPU[25]. Genes and exons with an average Transcripts Per Million (TPM) > 0.1 were considered to be expressed as described by Li et al. [53]. Population structure was controlled by the first three components from the expression principal components analysis. The resulting *p*-values were adjusted using Bonferroni correction at a level of α = 0.05. However, a potential issue of the FarmCPU model is its integration of leading markers as co-variants, which can remove co-expressed and result in false-negative TWAS results. To address this issue, TWAS was also conducted using a compressed mixed linear model implemented in GAPIT[54,55].

## DNA isolation and detection of *PH13* haplotype

The DNA of 1254 accessions (out of 2214 genome sequenced accessions) were extracted from leaves individually using the modified CTAB method[56]. The status of a SNP (A or G) located 2444 bp downstream of the translation start site (TSS) was determined using the resequencing data[24]. To identify the insertion of the *Ty1/Copia*-like retrotransposon, three primers were designed for polymerase chain reaction (PCR) detection (Supplementary Data 8). The presence of the fragment insertion was confirmed by agarose gel electrophoresis.

## Construction of near isogenic lines

The creation of a pair of near-isogenic lines (NILs), consisting of NIL[H1] and NIL[H3], was achieved through the crossbreeding of Williams82 carrying Hap1 (W82[H1]) and Tianlong1 carrying Hap3 (TL1[H3]). In the progenies of the F7 generation, a heterozygous line at the *PH13* gene was selected, and the F8 generation segregating groups, carrying homozygous H1 (NIL[H1]) and homozygous H3 (NIL[H3]) were used for phenotypic analysis (Supplementary Fig. 13). Segregation of the PH13 haplotypes was analyzed by PCR using the three primers designed for the *Ty1/Copia*-like retrotransposon insertion, and confirmed through agarose gel electrophoresis. Seeds heterozygous in the targeted region from the F8 generation were grown in the phytotron or field, and the plant height phenotypes of the segregating progeny were recorded at the R8 stage.

## Plasmid construction and plant transformation

To generate CRISPR/Cas9 engineered mutants, multiple gRNAs were designed for each gene on the basis of their specificity and off-target effects using the CRISPR direct website (http://crispr.dbcls.jp/)[57]. CRISPR/Cas9 vectors were constructed using multiple target gRNA for each gene in order to improve editing efficiency and minimize off-target effects[16]. The editing efficiency of each construct was evaluated using the soybean hairy root system[58], and at least two vectors independently with high editing efficiency for each gene were selected for soybean transformation. The selected CRISPR/Cas9 vectors were introduced into the *Agrobacterium tumefaciens* strain EHA105 through electroporation, and then separately transformed into W82[H1] and TL1[H3] using the cotyledon-node method[59].

To construct the overexpression vector, the coding DNA sequence (CDS) of *PH13* was amplified from cDNA derived from young W82[H1] seedlings via PCR. The CDS was then cloned into the *35 S::3×Flag* vector with an *XhoI* site, which was constructed based on the pFGC5941 plasmid containing the (2×35 S)-(3×Flag)-NOS cassette inserted between *EcoRI* and *HindIII* sites[60]. The newly generated construct, *35 S::PH13[H1]−3×Flag*, was then introduced into *Agrobacterium* strain EHA105 for the transformation of TL1[16]. Additionally, the *35 S::PH13[H1]−3×Flag* vector was introduced into the *Arabidopsis spa134* mutant[34] for ectopic expression. The applied primers were listed in the Supplementary Data 8.

## RNA extraction and gene expression analysis

To investigate the transcriptional dynamics of *PH13* in different genotypes and assess the co-expression of *PH13* and *GmCOP1s*, we grew W82[H1] and TL1[H3] separately under long-day conditions for 20 days. The second fully expanded trifoliolate leaves were harvested every 4 hours over a day. Two pairs of primers were designed for qPCR to detect the expression levels of different *PH13* alleles, one targeting exon 3, and the other targeting the 3′UTR. Total RNA was extracted using Trizol Reagent (TIANGEN), and then treated with DNase. A reverse transcription kit (TransGen Biotech) was used to synthesized cDNA from 3 μg of total RNA in a 20 μl reaction. RT-qPCR was performed on 384-well optical plate using the ABI Q7 equipment and the SYBR Green RT-PCR kit (Vazyme Biotech). All primers used for the indicated genes were listed in the Supplementary Data 8. Three independent biological replicates were carried out for each sample.

## Multiple alignment and phylogenetic analysis

The protein sequences of AtSPAs (AtSPA1 AT2G46340, AtSPA2 AT4G11110, AtSPA3 AT3G15354, AtSPA4 AT1G53090) were retrieved from TAIR (https://www.arabidopsis.org/index.jsp). Homologous SPA protein sequences were sourced from Phytozome (https://phytozome.jgi.doe.gov/pz/portal.html), with the list of the homologous SPA protein sequences utilized in this study outlined in Supplementary Data 4. The amino acid sequences of the SPA proteins and their homologous proteins were aligned by ClustalW in MEGA7 with manually adjustments. The phylogenetic tree was constructed using the neighbor joining method in MEGA7 software.

## Subcellular location in protoplasts

To investigate the subcellular location of different PH13 haplotype proteins, the CDSs of *PH13[H1]*, *PH13[H2]*, and *PH13[H3]* were inserted into the *pA7-YFP* vector at the *BamHI* and *SmaI* sites using the In-Fusion system (TransGen Biotech). The resulting *PH13[Hs]-YFP* transient expression constructs were driven by the *35 S* promoter. Control experiments were performed utilizing the empty *pA7-YFP* vector. Additionally the GmMYB29-RFP fusion protein was employed as a nucleus marker[61]. The plasmids were transformed into *Arabidopsis* mesophyll protoplasts and the resultant subcellular localization was imaged using a Zeiss LSM980 confocal laser scanning microscope. ZEN 2009 Light Edition software was used to process the images. All primers used for vector construction were listed in the Supplementary Data 8.

## Yeast two-hybrid assays

The yeast two-hybrid assays were conducted following the manufacturer's instructions (Yeast Handbook Clontech). The CDSs of *PH13[H1]*, *PH13[H2]*, *PH13[H3]*, *PH13N*, *PH13CT*, *PH13cc*, and *PH13WD40* were individually cloned in frame with the GAL4 DNA binding domain in the bait vector *pGBKT7* (Clontech, catalog no. 631604). While the CDSs of *GmCOP1a* and *GmCOP1b* were similarly fused with the GAL4 transcription activation domain in the prey vector *pGADT7* (Clontech, catalog no. K1612-1). The bait and prey plasmids were then co-transformed into the yeast strain *Saccharomyces cerevisiae* AH109 (Clontech). The transformed yeast cells were grown on SD/-Leu-Trp (-LW) minimal medium. Positive clones were selected and grown on SD/-His-Leu-Trp-Ade (-LWHA) selection medium at 30 °C for 3–5 days to evaluate protein interactions.

The β-galactosidase activity assay was performed as previously reported[62]. Briefly, yeast colonies were selected and cultured at 180 rpm and 28 °C in an incubator until the optical density (OD600) of the culture reached 0.1 in a 10 mL flask containing 4 ml of SD medium (-Leu/-Trp). 2 mL of yeast culture was then transferred into 8 mL YPDA culture solution and cultured at 180 rpm at 28 °C until the OD600 reached 0.5-0.8 prior to the β-galactosidase assay. The relative bait-prey interaction was presented as β-gal units (2), where T is the response time (min) and V represents 0.1 × concentration factor.

$$\beta - \text{gal units} = 1000 \times OD578/(T \times V \times OD600) \qquad (2)$$

## Co-immunoprecipitation assays

To evaluate the strength of interactions between different haplotypes of PH13 and GmCOP1b, a co-immunoprecipitation (Co-IP) assay was conducted in *N. benthamiana*. The *PH13[H1]-Flag*, *PH13[H2]-Flag* or *PH13[H3]-Flag* constructs were co-transformed with the *GmCOP1b-YFP* construct as indicated into *N. benthamiana* leaves. The YFP protein co-expressed with PH13[H1]Flag was used as a negative control. Following infiltration, the *N. benthamiana* plants were incubated at 25 °C for 12 h in the dark and then transferred to light growth conditions for an additional 36 h before IP analysis. The samples were harvested, ground, and treated with lysis buffer containing 1 mM MgCl$_2$, 10 mM EDTA [pH 8.0], 1 mM

PMSF, and 5 mM DTT, and Roche protease inhibitor cocktail. After centrifugation, the supernatant was incubated with anti-GFP trap agarose (Chromotek, catalog number gta-20) overnight at 4 °C, and rinsed three times with lysis buffer. The samples were boiled in SDS-PAGE sample buffer then analyzed using immunoblotting with anti-GFP antibody (at a 1:2500 dilution) followed by anti-Flag antibody (at a 1:2500 dilution).

## Light regimes

White light (WL), blue light (400–499 nm), red light (600–699 nm), and far-red light (700–750 nm) LED panels (HiPoint brand, 14005-11145, Made in TAIWAN) were used separately or in combination as indicated (Supplementary Fig. 24). Low blue light (LBL) was achieved by filtering WL through two layers of yellow filters (no. 101, Lee Filters, CA), while low red: far-red light (L R:FR) was achieved by supplementing WL with far-red light[21]. The height of the LED was adjusted to maintain the Photosynthetic Photon Flux Density (PPFD) at about 500 µmol m$^{-2}$ s$^{-1}$. The quality and intensity of light were measured by placing a HiPoint HR–350 spectrometer on top of the leaves.

## Statistical analyses

For phenotypic investigation, at least five individual plants per accession were analyzed. The exact numbers of individuals ($n$) varying depending on the experiment were presented in the figure legends. The expression analysis were conducted by pooling at least three individual plants per tissue sample and performing at least three RT-qPCR reactions (technical replicates) for three biological replicates. Multiple comparisons were performed using GraphPad Prism 8.0 software with a two-way ANOVA and a two-sided Tukey test. For comparisons between two groups, two-tailed Student's $t$-tests were conducted in Microsoft Excel to obtain $p$-values. The figure legends provide details on the statistical tests utilized for each experiment.

## Primers and accession number

All primers used in this study are listed in Supplementary Data 8. Gene sequences are available at the Phytozome database (https://phytozome-next.jgi.doe.gov/info/Gmax_Wm82_a2_v1): *PH13*/*Glyma*.13G276700, *PHP*/*Glyma*.12G224600, *GmCOP1a*/*Glyma*.02G267800, *GmCOP1b*/*Glyma*.14G049700, *STF1*/*Glyma*.08G302500, *STF2*/*Glyma*.18G117100, *GmCRY1a*/*Glyma*.04G101500, *GmCRY1b*/*Glyma*.06G103200, *GmCRY1c*/*Glyma*.14G174200 and *GmCRY1d*/*Glyma*.13G089200.

## Reporting summary

Further information on research design is available in the Nature Portfolio Reporting Summary linked to this article.

# Data availability

The raw sequence data[24] and gene expression data[63] reported elsewhere are available at NCBI Sequence Read Archive under accession PRJNA681974 and the Genome Sequence Archive (GSA) database of the BIG Data Center under accession PRJCA014188, respectively. The sequences of the three *PH13* alleles reported in this study are available at GenBank of NCBI: *PH13^{H1}*/OR637868, *PH13^{H2}*/OR637869, and *PH13^{H3}*/OR637870. Requests for materials should be addressed to B.L. or L.-J.Q. Source data are provided with this paper.

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

## Acknowledgements

This work was supported by the National Key Research and Development Plan (grant no. 2021YFF1001201, 2021YFD1201601), the National Natural Science Foundation of China (grant no. 31422041, 31871705, 32201759), the Innovation Program of Chinese Academy of Agricultural Sciences, the Agricultural Science and Technology Innovation Program (ASTIP) of the Chinese Academy of Agricultural Sciences (grant no. CAAS-ZDRW202109), the Central Public-Interest Scientific Institution Basal Research Fund, and the earmarked fund for CARS (grant no. CARS-04-PS01). We are grateful to Dr. Huihui Li (Institute of Crop Sciences, Chinese Academy of Agriculture Sciences) for the fruitful discussions.

## Author contributions

B.L., L.-J.Q., F.K. and Y.-H.L. designed the research. C.Q. and D.L. performed the experiments. X.L., R.J. and Q.C. provided relevant experimental materials. Y.Z., X.Z.W., Q.W., Y.W., W.H., Q. Z., L.L., X.W., G.X., G.H., Z.S. and R.W., assisted in collecting the phenotypic data. Z.J., X.J.L., Y.L., H.L., H.Y.L., T.Z., J.L. and X.H designed the field experiment. C.Q., X.Z., D.L. and L.K. analyzed data. B.L., L.-J.Q., F.K., Y.-H.L., D.L. and C.Q. wrote the manuscript.

## Competing interests

The authors declare no competing interests.
