## [Peer Review File · Nature Communications]

PH13 improves soybean shade traits and enhances yield for high-density planting at high latitudesReviewers' Comments:

Reviewer #1:

Remarks to the Author:

What are the noteworthy results?

Qin et al. present an interesting study looking at genes that affect plant height in soybean. They found three alleles for a WD40 protein on chromosome 13 that appears to correlate with plant height. A transposon insertion in the coding region of one of the alleles - PH13(H3) - causes plants to have shorter internodes. They report that this allele is more frequently found in northern latitudes, leading them to conclude that it is important for reducing lodging in the long-day, short season environments. They demonstrate that the wild type allele encodes a protein that interacts with GmCOP1, while PH13(H3) does not interact with this pathway, leading to accumulation of STF1/2 and reducing height. They then use gene editing to create a double-mutant with the paralog of PH13(H3). This double mutant (named phd) does not show stem elongation under shaded light conditions. They conclude that phd may be useful for planting in shading conditions, such as intercrop with maize, because it will have reduced lodging.

Will the work be of significance to the field and related fields? How does it compare to the established literature?

In my opinion, this work will be significant to the plant biology community, particularly those working in crop systems that are interested in understanding the genes that underlie above-ground architecture traits.

Does the work support the conclusions and claims, or is additional evidence needed?

The main conclusion of this work is that the genes identified are important for soybeans grown in northern latitudes and/or intercropping systems. I am not entirely convinced that the PH13(H3) was selected by breeders for these purposes, but I think there is compelling evidence to form this hypothesis. In any case, I think the identification of the alleles and pathway affected and their relationship to the architecture traits are significant findings.

Are there any flaws in the data analysis, interpretation and conclusions? Do these prohibit publication or require revision?

I did not see any fatal flaws in the analysis, interpretation, or conclusions in this manuscript.

Is the methodology sound? Does the work meet the expected standards in your field?

The methodology appears to be sound. While I do not like soybean architecture studies to be performed on growth chamber and/or greenhouse grown plants, this manuscript also includes field phenotyping of their genetic materials to support their conclusions.

Is there enough detail provided in the methods for the work to be reproduced?

Yes.

Additional major comments:

-The authors demonstrate that phd double-mutant plants are shorter and have thicker stems. They conclude that these are good "shade tolerance" traits because they don't grow taller in response to shading; this is nicely demonstrated in Supplementary Figure 23. However, the phd plants do not show a grain yield advantage in the intercropping system compared to traditional varieties. Could it be that short pdh plants receive less sunlight when intercropped with maize? And this would explain the relative lack of yield advantage of these plants? Or do the authors think the grain yield is similar between the lines because the genetic background of the phd plants is inferior to the traditional lines? I think the authors should discuss this topic a bit in the Discussion section.

-"Of these, fourteen have not been reported previously as soybean height loci..." The authors should

be aware that this region on chromosome 13 has been implicated in soybean height QTL in the past. Three relatively recent papers include Oki et al. (Breed Sci. 68: 554–560), Diers et al. (G3 8: 3367–3375), and Wang et al. (Plant J 107: 1739–1755). I am not sure if these are the same QTL as reported here by Qin et al., but the authors may want to look at these papers to check. If they are not the same QTL, the authors still may want to cite these papers to clarify the distinction. I also encourage the authors to double-check the literature to see if any of the other 13 QTL have been previously been reported as height QTL.

-Line 288: “different planting densities (30 cm, 20 cm, 10 cm, and 5 cm plant space equivalent to...” And later (line 376): “soybean was spaced 5 cm apart in 5 m long rows”. How was the 5 cm seeding distance done? Was it done by machine planting, hand planting, or seedling transplant. Please specify. Were stand counts performed to estimate the frequency of failed emergence for the different genotypes?

-The authors show much data about the plant height and architecture of their mutants. However, they often fail to mention where the plants were grown (field, growth chamber, greenhouse, etc.). I have mentioned several examples in the Minor comments, but the authors should be sure to do this for every instance where they show phenotypes (in the Results text, the figure legends, or both). As we all know, soybean architecture traits respond very strongly to the local growing environment, so this is an important point.

Minor comments:

-Line 54: “providing 56% edible oil, 25% protein and other nutritional resources...” What is the context for these numbers? Normally, it is said that soybean seed consists of ~40% protein and ~20% oil. So, I think the authors need to provide more context for their numbers, to make sure readers are not confused.

-Line 93: “ideatype” should be “ideal type”

-Line 143: Please state whether these plants were grown in the greenhouse, growth chamber, or the field. If in the field, please state the location.

-Line 156: Please state whether these plants were grown in the greenhouse, growth chamber, or the field. If in the field, please state the location.

-Line 203: Please state whether the sampled plants were grown in the greenhouse, growth chamber, or the field. If in the field, please state the location.

-Line 228: Please state whether the sampled plants were grown in the greenhouse, growth chamber, or the field. If in the field, please state the location.

-Line 248: “act upstream regulators” should be “act as upstream regulators”

-Line 262: Please define “SAS”

-Line 267: “number in compared with TL1H3” should be “number in comparison with TL1H3”

-Line 326: Change “unconsciously” to “unintentionally”

-Figure 1c legend: The terminology “zero-expressed” and “non-zero-expressed” is confusing. Can you use more accessible terminology? Or just use the allele names?

-Figure 1d legend: Explain what the red “G” (base 2444) indicates.

-Figure 1f legend: Please state whether these plants were grown in the greenhouse, growth chamber, or the field. If in the field, please state the location.

Figure 2a legend: Please specify the "natural conditions..." Does this mean the field? If so, state "natural field conditions..."

-Figure 2c legend: Please state whether these plants were grown in the greenhouse, growth chamber, or the field. If in the field, please state the location.

- Figure 4a-c legend: Please state whether the sampled plants were grown in the greenhouse, growth chamber, or the field. If in the field, please state the location.

- Figure 4f-h legend: Please state whether the sampled plants were grown in the greenhouse, growth chamber, or the field. If in the field, please state the location.

-Figure 21a: What is "Height of gravity"? Please define.

-Supplementary Table 5: As with all the files, this was converted to pdf for review. It seems the amino acid sequences are only partially viewable in the pdf format. If this gets published, this may need to be made available as a spreadsheet file.

Reviewer #2:

Remarks to the Author:

This is an exciting paper that uncovers the molecular/genetic basis of variation in an agronomically important trait in soybean. It has previously been shown that soybean grown at high latitudes suffers from excessive elongation. Here GWAS and TWAS were used to identify a new locus that plays an important role in controlling this trait. The authors provide convincing evidence that they have identified the correct gene, a soy homolog of the known Arabidopsis light signaling SPA gene family. Furthermore, they go on to create and then show that knock out mutations of this gene and its paralog may create lines that may be even better suited to high-latitude growth. The authors further use genetic epistasis and protein-protein interaction analyses to provide insight into where this gene acts in the light signaling pathway (consistent with what is known from Arabidopsis) and how the natural variant alters its interaction with the downstream signaling partner COP1. Writing is clear, analysis is strong. There is a brief analysis showing allele frequency change over time and the geographic distribution of the variant allele, consistent with it having been selected for in elite varieties bred for growth in high latitudes. While there is no analysis of genomic signatures of selection, I do not think that type of analysis is needed for this paper, given the strength and breadth of the genetic and molecular work.

A few minor comments:

1) line 118 "homology to the [...] SPA protein". Vague. There are four SPA proteins in Arabidopsis. Either say "to the SPA family of proteins" or note that it specifically seems to be a member of the SPA3/4 group.

2) Phylogenetic analysis. It is very unlikely to change the result, but neighbor joining is an outdated procedure. Maximum likelihood would be a better choice. I do not consider this critical for this paper since it is clear that this is a SPA homolog and the phylogenetic relationship is not the main point of the paper. Still it would be nice...

3) line 226 "responsible for stem elongation". This is backwards or at least confusing. The wild-type

function of these genes is to inhibit stem elongation.

4) line 333. SPA1 is referred to as "the homologous protein of PH13" but this is incorrect based on the presented phylogenetic tree. PH13 is more closely related to SPA3/4.

5) line 395. More information should be given about the statistical model. Was this a mixed-effect model? What were the fixed and random effects? How was year handled? Were there any interaction terms?

6) It would be helpful for the methods to reference the supplemental figure that shows the various light spectra used in the controlled environment experiments.

minor word choice suggestions:

line 70 "north of America"  "northern United States" (since Canada is listed separately I assume that in this case "America" really means "United States")

line 239 "another"  "a"

line 247 "than in"  "compared to"

line 296 "remained"  "remaining"

line 326 "unconsciously"  "unknowingly"

Reviewer #3:

Remarks to the Author:

The manuscript by Qin and coworkers describes the role of locus PH13 in soybean, which shows homology to Arabidopsis SPA3. The role of PH13 in regulating plant height was discovered through genome-wide association studies. PH13 exists as 3 haplotypes and the soy plants expressing a WD40-truncated version of PH13 (PH13H3), which is unconsciously selected agronomically, show shade-tolerant characteristics under high altitudes. The authors propose that the deletion of a major part of WD40 in PH13 reduces its interaction with GmCOP1 and thus results in enhanced accumulation of STF (homologous to HY5 in Arabidopsis).

The experiments are well conducted and the results are presented appealingly. I appreciate the meticulous efforts by the authors in performing many of the time-taking and tedious experiments described in the manuscript. At the same time, I have the following suggestions that would greatly improve the manuscript.

1. As shown in Figure 4e, PH13H3 interacts weakly with GmCOP1s and the authors argue that PH13 interaction with GmCOP1 may not be via their CC domain (lines 333-334). However, the evidence for this scenario was not presented. It is equally likely that the truncated WD40 in PH13H3 do not fold properly or engage in confirmation with the CC domain that may prevent GmCOP1 interaction. I suggest testing this hypothesis by performing domain-specific Y2H assays in which the interaction between N-terminal, CC and WD40 domains with GmCOP1s are tested individually

2. Figure 4h, suppl. Figure 12d: The differences in the total protein amounts in different samples, reflected by the differences in HSP70 have not been included in the calculation. This is especially important since the HSP70 tends to be higher in NILH13 samples and low in the 24 h samples on the bottom panel of the suppl. Figure 12c. How many replicate experiments were performed for individual blots presented in the manuscript? I would suggest doing at least 2-3 replicates for each blot and then calculating the average before plotting the data (applicable to all the figures with immunoblots)

3. Nuclear localization of PH13: Suppl. Figure 5b shows the localization in the nucleus as well as in the cytoplasm. From the figure, they are distributed throughout and there are no dramatic differences in nuclear localization. This is especially peculiar since the nuclear localization of COP1 is necessary for its activity. I would suggest including these points in the discussion section
4. The TWAS RNA-seq data should be uploaded to a public database to give access to reviewers and readers
5. In Figure 1b, the legend does not provide any information about the green and orange colours used in the Manhattan plot. Also, apart from PH13, TWAS in Figure 1b shows at least two additional genes (orange colour)
6. qRT-PCR is not the right word. It should be RT-qPCR as it is not quantitative in real-time. This should be changed throughout the manuscript
7. Suppl. Figure 12c: Do the asterisks indicate non-specific binding of the anti-STF antibody?
8. Data points in suppl. Figure 21d and e, suppl. Figure 22d and e: They can be shown scattered like the other plots in the same figure
9. Line 264: Please give reference for GmCRY1-qm mutant. Also, indicate how the phd-1/GmCRY1-qm line was made
10. Line 79: Change "of" to "in"
11. Apart from PH13 and PHP, there are two more proteins GmSPA3c and GmSPA3d. The authors may include them in the alignment data shown in suppl. Figure 4

Reviewer #4:

Remarks to the Author:

This manuscript starts out from a GWAS study for plant height in soybean, combined with TWAS for the same. A locus is identified from this study (PH13), which is then studied in further detail. The authors make the case that PH13 is homologous to Arabidopsis SPA genes, which work in a complex with COP1 towards protein degradation. The wrap up with showing that engineering PH13 affects soybean yield.

The manuscript is well written and has a logical flow. It does take a very similar path as a previous paper from some of the same authors published in Molecular Plant a few years ago, that studies components of the same regulatory pathway towards plant height in soybean.

In your manuscript you conclude that PH13 is a SPA-homologue, acting together with COP1 to regulate stability of STF1/2, which are HY5 homologues. This may be true, and confirms exactly what is very well established already in Arabidopsis. My concern is, therefore, that your study is rather confirmatory of Arabidopsis knowledge, now in soybean.

You do take this further to show that yield is affected, but this was shown previously already for soybean STF1/2 and CRY manipulation; essentially the pathway you study here as confirmed in your current data also.

In your GWAS and TWAS you decided to focus on PH13 since this was the locus peak that overlapped between both. However, several other peaks showed up, some even with stronger scores. It would be relevant to explain more about their identity. A table with all significant peaks, their identity, gene family and function(if known) would be very helpful.

I am not sure that Figure 3 really has to be a main figure, it would do well as a supplemental figure.

You conclude, based on geographical distribution of the different PH13 alleles and knowledge on breeding patterns, that this gene is causally related to selection in soy breeding. It would be important for this claim to be supported by proof around this specific locus. For example, bringing the PH13H1 allele into a modern, dwarfed variety that carries the PH13H3 allele (with its insertion), should restore plant height.

Causality: I am missing introgression of PH13H1 into PH13H1-expressing cultivars to confirm that really this locus is causal to the height and yield phenotypes of such cultivars.

To corroborate your interpretation of PH13 being a SPA-like protein, it would be useful to complement a PH13H3 soybean cultivar with a functional SPA homologue from Arabidopsis. Likewise, you clone PH13H1 and express it in Arabidopsis spa mutants to verify if these are truly homologues.

In Fig 6 you draw root systems, but you have not looked into these. Therefore, any suggestion about root development should be avoided. This is quite relevant since HY5 and presumably STF1/2 in soybean, might control root architecture.

Lines 270-272: You did not study shade tolerance, and can therefore also not draw the conclusion on "excellent shade-tolerance traits".

Reply to referees:

We appreciate the constructive feedback provided by the four reviewers and the editor. We have carefully reviewed and revised our manuscript in response to their valuable comments and suggestions. We have taken the opportunity to perform additional experiments, including domain-specific Y2H assays, replication for each immunoblot, transformation experiments, and provided necessary planting information and new discussion points. Please see our point-by-point response below.

#####

REVIEWER COMMENTS

Reviewer #1 (Remarks to the Author):

What are the noteworthy results?

Qin et al. present an interesting study looking at genes that affect plant height in soybean. They found three alleles for a WD40 protein on chromosome 13 that appears to correlate with plant height. A transposon insertion in the coding region of one of the alleles - PH13(H3) - causes plants to have shorter internodes. They report that this allele is more frequently found in northern latitudes, leading them to conclude that it is important for reducing lodging in the long-day, short season environments. They demonstrate that the wild type allele encodes a protein that interacts with GmCOP1, while PH13(H3) does not interact with this pathway, leading to accumulation of STF1/2 and reducing height. They then use gene editing to create a double-mutant with the paralog of PH13(H3). This double mutant (named phd) does not show stem elongation under shaded light conditions. They conclude that phd may be useful for planting in shading conditions, such as intercrop with maize, because it will have reduced lodging.

Will the work be of significance to the field and related fields? How does it compare to the established literature?

In my opinion, this work will be significant to the plant biology community, particularly those working in crop systems that are interested in understanding the genes that underlie above-ground architecture traits.

Does the work support the conclusions and claims, or is additional evidence needed?

The main conclusion of this work is that the genes identified are important for soybeans grown in northern latitudes and/or intercropping systems. I am not entirely convinced that the PH13(H3) was selected by breeders for these purposes, but I think there is compelling evidence to form this hypothesis. In any case, I think the identification of the alleles and pathway affected and their relationship to the architecture traits are significant findings.

Are there any flaws in the data analysis, interpretation and conclusions? Do these prohibit publication or require revision?

I did not see any fatal flaws in the analysis, interpretation, or conclusions in this manuscript.

Is the methodology sound? Does the work meet the expected standards in your field?

The methodology appears to be sound. While I do not like soybean architecture studies to be performed on growth chamber and/or greenhouse grown plants, this manuscript also includes field phenotyping of their genetic materials to support their conclusions.

Is there enough detail provided in the methods for the work to be reproduced?

Yes.

Additional major comments:

Comment 1:

-The authors demonstrate that *phd* double-mutant plants are shorter and have thicker stems. They conclude that these are good “shade tolerance” traits because they don’t grow taller in response to shading; this is nicely demonstrated in Supplementary Figure 23. However, the *phd* plants do not show a grain yield advantage in the intercropping system compared to traditional varieties. Could it be that short *phd* plants receive less sunlight when intercropped with maize? And this would explain the relative lack of yield advantage of these plants? Or do the authors think the grain yield is similar between the lines because the genetic background of the *phd* plants is inferior to the traditional lines? I think the authors should discuss this topic a bit in the Discussion section.

Reply: Thank you for your suggestions. The control lines LK317 and LK18-842 are considered to be among best elite cultivars for both normal planting and maize-soybean relay intercropping in the norther region of China. As such, we prioritize that the *phd* plants do not show a grain yield advantage compared to the control lines LK317 and LK18-842, largely because the genetic background of the *phd* plants is inferior to that of the two control lines. To further elaborate, we have added additional sentences in the discussion section (Line 374-380):

“Moreover, the phd mutations can also benefit intercropping by reducing the lodging-induced yield reduction (Supplementary Fig. 30d-e). Although the phd mutant in the TL1 background exhibited lower yield potential compared to the elite cultivars LK317 and LK18-842 in Northern China under norm planting conditions, it displayed a lower lodging rate and a similar grain yield potential relative to the elite cultivars LK317 and LK18-842 under maize-soybean relay intercropping conditions.”

Comment 2:

-“Of these, fourteen have not been reported previously as soybean height loci...” The authors should be aware that this region on chromosome 13 has been implicated in soybean height QTL in the past. Three relatively recent papers include Oki et al. (Breed Sci. 68: 554–560), Diers et al. (G3 8: 3367–3375), and Wang et al. (Plant J 107: 1739-1755). I am not sure if these are the same QTL as reported here by Qin et al., but the authors may want to look at these papers to check. If they are not the same QTL, the authors still may want to cite these papers to clarify the distinction. I also encourage the authors to double-check the literature to see if any of the other 13 QTL have been previously been reported as height QTL.

Reply: We apologize for not including the relevant literature in our manuscript and thank you for your valuable suggestion to improve it. The three papers you recommended have identified plant height QTLs on Chr13 that are either adjacent or encompass the *PHI3* gene (*Glyma.13g276700*, Gm13 37813197-37819453) (Diers et al., G3, 2018; Wang et al., Plant Journal, 2021; Oki et al., Breeding Science, 2018). Additionally, we have found two other studies that have reported similar QTLs (Satt490, Kabelka et al., 2004; Satt090, Yarmilla et al., 2006), which are 4.6 Mb and 1.1 Mb away from *PHI3*, respectively. These findings suggest that the region surrounding the *PHI3* gene is a significant locus for controlling plant height. To further elaborate, we have added additional sentence in the main text (Line 115-117):

“The surrounding genetic region of PHI3 has repeatedly been identified as a plant height-associated QTL in previous studies²⁸⁻³², supporting a role of PHI3 in regulating plant height.”

Comment 3:

-Line 288: “different planting densities (30 cm, 20 cm, 10 cm, and 5 cm plant space equivalent to...” And later (line 376): “soybean was spaced 5 cm apart in 5 m long rows”. How was the 5 cm seeding distance done? Was it done by machine planting, hand planting, or seedling transplant. Please specify. Were stand counts performed to estimate the frequency of failed emergence for the different genotypes?

Reply: Thanks for your question. We have included the planting information in the Online Methods section (Lines 410-414):

“To ensure consistency, all lines under different densities were manually planted with 3 seeds per hole. Once the seeds germinated and the unifoliate leaf fully unfolded, only one healthy seedling was retained per hole. Any failed seedlings were removed. To maintain uniformity in each planting density, empty holes were filled with a healthy seedling of same genotype through transplanting.”

Comment 4:

-The authors show much data about the plant height and architecture of their mutants. However, they often fail to mention where the plants were grown (field, growth chamber, greenhouse, etc.). I have mentioned several examples in the Minor comments, but the authors should be sure to do this for every instance where they show phenotypes (in the Results text, the figure legends, or both). As we all know, soybean architecture traits respond very strongly to the local growing environment, so this is an important point.

Reply: We apologize for the oversight in not including the detailed growth conditions for plants displaying phenotypes in previous versions of the article. We understand that soybean architecture traits are highly influenced by the local growing environment, and we have made sure to include the necessary planting information for all materials in this article. Please refer to our responses to the following Minor comments for further details.

Minor comments:

1. -Line 54: “providing 56% edible oil, 25% protein and other nutritional resources...” What is the context for these numbers? Normally, it is said that soybean seed consists of ~40% protein and

~20% oil. So, I think the authors need to provide more context for their numbers, to make sure readers are not confused.

Reply: Thank you for bringing this mistake to our attention. Our intention was to highlight the significance of soybean in providing large quantities of plant oil and protein in the world. However, we apologize for any confusion caused by inaccurately conveying this information. We have carefully reviewed the latest statistical data and revised the sentence in the main text (Line 54-56) to accurately reflect the current situation:

*“Soybean (*Glycine max* (L.) Merr.) is an economically important crop, accounting for 59% of the oilseed production and providing 70% of the plant protein for human and animal consumption worldwide (SoyStats, 2023).”*

2. -Line 93: “ideatype” should be “ideal type”

Reply: Revised.

3. -Line 143: Please state whether these plants were grown in the greenhouse, growth chamber, or the field. If in the field, please state the location.

Reply: Thank you for your suggestion to clarify the planting conditions for the plants. We have added detailed information on the planting conditions in the manuscript as follows:

Main text (Line 150-153):

“Moreover, PH13^{H3} accessions were found to have significantly reduced plant height compared to accessions carrying PH13^{H1} and PH13^{H2} under field conditions in ten field locations over two or three years (Fig. 1f), suggesting that the retrotransposon insertion in PH13^{H3} is responsible for the reduction in plant height.”

Figure 1f legend (Line 787-790):

“Distribution of plant height BLUP (Best Linear Unbiased Prediction) for each haplotype. The BLUP values were calculated using the plant height data of natural populations in ten field environments over two or three years (Online Methods, GWAS and TWAS assays).”

4. -Line 156: Please state whether these plants were grown in the greenhouse, growth chamber, or the field. If in the field, please state the location.

Reply: We added the detailed information in the Figure 2a legend (Line 796-797):

“Gross photos of the indicated lines grown under natural field conditions in the summer of Beijing.”

5. -Line 203: Please state whether the sampled plants were grown in the greenhouse, growth chamber, or the field. If in the field, please state the location.

Reply: We added the detailed information in the Figure 4c legend (Line 823-825):

“The second trifoliolate leaves of 20-day-old seedlings grown under long-day conditions in growth chamber were collected at 4-hour intervals for RT-qPCR analysis.”

6. -Line 228: Please state whether the sampled plants were grown in the greenhouse, growth chamber, or the field. If in the field, please state the location.

Reply: We added the detailed information in the Figure 4g legend (Line 838-840):

“The first trifoliolate leaves of 15-day-old seedlings grown under long-day condition in growth chamber were collected at 4-hour intervals.”

7. -Line 248: “act upstream regulators” should be “act as upstream regulators”

Reply: Revised.

8. -Line 262: Please define “SAS”

Reply: Revised as follows (Line 276-277):

“The blue light receptor, GmCRY1s, was observed to mainly mediated LBL-induced shade avoidance syndrome (SAS) in soybean.”

9. -Line 267: “number in compared with TL1H3” should be “number in comparison with TL1H3”

Reply: Revised.

10. -Line 326: Change “unconsciously” to “unintentionally”

Reply: Revised.

11. -Figure 1c legend: The terminology “zero-expressed” and “non-zero-expressed” is confusing. Can you use more accessible terminology? Or just use the allele names?

Reply: Thank you for your suggestion, and we apologize for any confusion caused by our previous description. We have replaced the confusing terminology with detail descriptions in the Figure 1c legend to ensure clarity. Specifically, we have made the following changes (Line 775-779):

“Two types of transcripts associated with plant height variation were shown. The transcripts with normal expression levels are displayed at the top, while those with truncated expression due to the insertion site of a Ty1/Copia-like retrotransposon are shown below. Five samples from each category were randomly selected and pooled for alignment visualization.”

12. -Figure 1d legend: Explain what the red “G” (base 2444) indicates.

Reply: We thank the reviewer for raising this question, and added description in the Figure 1d legend (Line 782-783):

“The red “G” (base 2444) represents the nonsynonymous mutation derived from exonic SNP.”

13. -Figure 1f legend: Please state whether these plants were grown in the greenhouse, growth chamber, or the field. If in the field, please state the location.

Reply: We added the detailed information in the Figure 1f legend (Line 787-790):

“Distribution of plant height BLUP (Best Linear Unbiased Prediction) for each haplotype. The BLUP values were calculated using the plant height data of natural populations in ten field environments over two or three years (Online Methods, GWAS and TWAS assays).”

14. -Figure 2a legend: Please specify the “natural conditions...” Does this mean the field? If so, state “natural field conditions...”

Reply: Thank you for your suggestion. We added the detailed information in the Figure 2a legend (Line 796-797):

“Gross photos of the indicated lines grown under natural field conditions in the summer of Beijing.”

15. -Figure 2c legend: Please state whether these plants were grown in the greenhouse, growth chamber, or the field. If in the field, please state the location.

Reply: We added the detailed information in the Figure 2c legend (Line 801-803):

“Plant height of the near-isogenic lines (NILs) carrying homozygous H1 (NIL^{H1}) and homozygous H3 (NIL^{H3}) under natural field conditions in Beijing.”

16. -Figure 4a-c legend: Please state whether the sampled plants were grown in the greenhouse, growth chamber, or the field. If in the field, please state the location.

Reply: We added the detailed information in the Figure 4a-c legend (Line 823-825):

“The second trifoliolate leaves of 20-day-old seedlings grown under long-day conditions in growth chamber were collected at 4-hour intervals for RT-qPCR analysis.”

17. -Figure 4f-h legend: Please state whether the sampled plants were grown in the greenhouse, growth chamber, or the field. If in the field, please state the location.

Reply: We added the detailed information in the Figure 4f-h legend:

Line 832 *“The transformed plants were incubated at 25°C in dark for 12 h and then grown under white light (WL, 80 $\mu\text{mol m}^{-2} \text{s}^{-1}$) for 36 h in growth chamber.”*

Line 838 *“The first trifoliolate leaves of 15-day-old seedlings grown under long-day condition in growth chamber were collected at 4-hour intervals.”*

18. -Figure 21a: What is “Height of gravity”? Please define.

Reply: We apologize for not making it clear. We measured the height of the equilibrium point at which mature plants were placed in horizontal arrangement, which we considered to be the height

of the center of gravity point and a key factor in determining soybean lodging resistance. For clarity, we have changed 'Height of gravity' to 'Height of the center of gravity point' in Line 418 and Supplementary Fig. 28, and defined it as follows (Line 419-421):

“The height of the center of gravity point is a crucial determinant of soybean lodging resistance, which is measured as the height of the equilibrium point at which mature plants were placed in horizontal arrangement.”

19. -Supplementary Table 5: As with all the files, this was converted to pdf for review. It seems the amino acid sequences are only partially viewable in the pdf format. If this gets published, this may need to be made available as a spreadsheet file.

Reply: Thank you for bringing this to our attention. We have updated the Supplementary Table 5 to provide all data in a spreadsheet file format.

Reviewer #2 (Remarks to the Author):

This is an exciting paper that uncovers the molecular/genetic basis of variation in an agronomically important trait in soybean. It has previously been shown that soybean grown at high latitudes suffers from excessive elongation. Here GWAS and TWAS were used to identify a new locus that plays an important role in controlling this trait. The authors provide convincing evidence that they have identified the correct gene, a soy homolog of the known Arabidopsis light signaling SPA gene family. Furthermore, they go on to create and then show that knock out mutations of this gene and its paralog may create lines that may be even better suited to high-latitude growth. The authors further use genetic epistasis and protein-protein interaction analyses to provide insight into where this gene acts in the light signaling pathway (consistent with what is known from Arabidopsis) and how the natural variant alters its interaction with the downstream signaling partner COP1. Writing is clear, analysis is strong. There is a brief analysis showing allele frequency change over time and the geographic distribution of the variant allele, consistent with it having been selected for in elite varieties bred for growth in high latitudes. While there is no analysis of genomic signatures of selection, I do not think that type of analysis is needed for this paper, given the strength and breadth of the genetic and molecular work.

Reply: Thank you very much for acknowledging our efforts in identifying and characterizing the *PHI3* gene in soybean. We believe it is a great idea to analyze the genomic signatures of selection in the *PHI3* gene, and we will try to perform this analysis in the future. Here are our point-to-point responses to your questions for your convenience to review.

A few minor comments:

1) line 118 "homology to the [...] SPA protein". Vague. There are four SPA proteins in Arabidopsis. Either say "to the SPA family of proteins" or note that it specifically seems to be a member of the SPA3/4 group.

Reply: Thank you for your suggestion. We revised this sentence as follows (Line127-129):

"The PHI3 gene encoding a WD40 protein which is homologous to the suppressor of the phyA-105 (SPA) family protein in Arabidopsis. Phylogenetic analysis indicated that PHI3 is grouped closely with SPA3/4 protein (Supplementary Fig. 2a)."

2) Phylogenetic analysis. It is very unlikely to change the result, but neighbor joining is an outdated procedure. Maximum likelihood would be a better choice. I do not consider this critical for this paper since it is clear that this is a SPA homolog and the phylogenetic relationship is not the main point of the paper. Still it would be nice...

Reply: Thanks for your suggestion. We have performed the phylogenetic analysis by Maximum Likelihood method, and update the phylogenetic analysis in Supplementary Fig. 2a.

Supplementary Fig. 2a: Phylogenetic tree of SPA family members.

Phylogenetic tree of SPA family members from *Arabidopsis thaliana*, rice, and soybean. The tree was constructed using the Maximum Likelihood method in the MEGA7 software.

3) line 226 "responsible for stem elongation". This is backwards or at least confusing. The wild-type function of these genes is to inhibit stem elongation.

Reply: Thank you for pointing out this issue. We have revised this sentence in the main text (Line 238-240):

“Given previous evidence that GmCOP1s mediate the degradation of STF1/2 transcription factors which are homologous to Arabidopsis HY5 and responsible for inhibiting stem elongation in legume.....”

4) line 333. SPA1 is referred to as "the homologous protein of PH13" but this is incorrect based on the presented phylogenetic tree. PH13 is more closely related to SPA3/4.

Reply: Thank you for pointing out this question. We have revised this sentence in the main text (Line 353-356):

“This is likely different from Arabidopsis, where the interaction between SPA1 (a homologous protein of SPA3/4 and PH13) and COP1 is also mediated by the coil-coil domains, but the absence of the WD40 domain of SPA1 does not impair the interaction with COP1.”

5) line 395. More information should be given about the statistical model. Was this a mixed-effect model? What were the fixed and random effects? How was year handled? Were there any interaction terms?

Reply: We apologize for not providing the necessary details earlier. Our study used a mixed-effects model to analyze the relationship between soybean plant height and various factors such as line, location, and year. The fixed effect in this model was the overall mean plant height across all

lines, locations, and years. The random effects accounted for variations within each line, location, year, and their interactions including Line: Location and Line: Year. We chose this approach to handle missing data, as some lines did not have a recorded plant height value in certain locations or years due to soybean photoperiod sensitivity and other environmental factors. To ensure clarity and reproducibility, we have revised the Methods section (lines 435-443) as follows:

“The soybean lines’ plant height BLUPs (Best Linear Unbiased Predictions) were calculated using a mixed linear model (E1). In this model, Y represents the observed plant height, X is the fixed effects, β is the vector of fixed effect coefficients, Z is the random effects, u is the random effect coefficients, e is the residual errors.

$$Y=X\beta+Zu+e \text{ (E1)}$$

The fixed effect in the model is the overall mean plant height across all lines, locations, and years. Random effects account for variations within each line, location, year, and their interactions (Line: Location and Line: Year). To account for the missing data in some lines across experimental trials, both Year and Location were treated as random effects.”

6) It would be helpful for the methods to reference the supplemental figure that shows the various light spectra used in the controlled environment experiments.

Reply: Thanks for your suggestion. We have added the “Light Regimes” in Online Methods (Line 574-581) as follows:

“White light (WL), blue light (400-499 nm), Red light (600-699 nm), and Far-red light (700-750 nm) LED panels (HiPoint brand, Z800000001) were used separately or in combination as indicated (Supplementary Fig. 20). Low blue light (LBL) was achieved by filtering WL through two layers of yellow filters (no. 101, Lee Filters, CA), while low red: far-red light (L R:FR) was achieved by supplementing WL with far-red light as described previously²¹. By adjusting the height of the LED, the Photosynthetic Photon Flux Density (PPFD) is maintained at about 500 $\mu\text{mol m}^{-2} \text{s}^{-1}$. The light quality and intensity were measured using a HiPoint HR-350 spectrometer.”

minor word choice suggestions:

1). line 70 "north of America"  "northern United States" (since Canada is listed separately I assume that in this case "America" really means "United States")

Reply: Thanks for your suggestion. We revised "north of America" to "northern United States".

2). line 239 "another"  "a"

Reply: Revised.

3). line 247 "than in"  "compared to"

Reply: Revised.

4). line 296 "remained"  "remaining"

Reply: Revised.

5). line 326 "unconsciously"  "unknowingly"

Reply: Revised.

Reviewer #3 (Remarks to the Author):

The manuscript by Qin and coworkers describes the role of locus PH13 in soybean, which shows homology to Arabidopsis SPA3. The role of PH13 in regulating plant height was discovered through genome-wide association studies. PH13 exists as 3 haplotypes and the soy plants expressing a WD40-truncated version of PH13 (PH13H3), which is unconsciously selected agronomically, show shade-tolerant characteristics under high altitudes. The authors propose that the deletion of a major part of WD40 in PH13 reduces its interaction with GmCOP1 and thus results in enhanced accumulation of STF (homologous to HY5 in Arabidopsis).

The experiments are well conducted and the results are presented appealingly. I appreciate the meticulous efforts by the authors in performing many of the time-taking and tedious experiments described in the manuscript. At the same time, I have the following suggestions that would greatly improve the manuscript.

1. As shown in Figure 4e, PH13H3 interacts weakly with GmCOP1s and the authors argue that PH13 interaction with GmCOP1 may not be via their CC domain (lines 333-334). However, the evidence for this scenario was not presented. It is equally likely that the truncated WD40 in PH13H3 do not fold properly or engage in confirmation with the CC domain that may prevent GmCOP1 interaction. I suggest testing this hypothesis by performing domain-specific Y2H assays in which the interaction between N-terminal, CC and WD40 domains with GmCOP1s are tested individually

Reply: We greatly appreciate for your constructive suggestion. To test if the interaction between PH13 and GmCOP1 is mediated through their cc domains, we performed domain-specific Y2H assays and β -galactosidase assays. The results showed that PH13 does interact with GmCOP1b via their coil-coil domain, but the WD40 domain can enhance their interaction strength. The results are included in the revised manuscript (Line 231-235, Supplementary Fig. 13) as follows:

“Domain-specific Y2H assays indicated that PH13 interacts with GmCOP1b via their coil-coil domains, but the WD40 domain of PH13 can significantly enhance their interaction strength (Supplementary Fig. 13a-c). These results together demonstrate that the absence of the WD40 domain in PH13^{H3} reduces the interaction strength between PH13 and GmCOP1s in soybean.”

The related contents were also revised in the discussion section (Line 352-356):

“The interaction between PH13 and GmCOP1b is mediated by their coil-coil domains, and enhanced by the WD40 domain of PH13. This is likely different from Arabidopsis, where the interaction between SPA1 (a homologous protein of SPA3/4 and PH13) and COP1 is also mediated by the coil-coil domains, but the absence of the WD40 domain of SPA1 does not impair the interaction with COP1.”

Supplementary Fig. 13: Auxotrophic assays showing the interactions between different haplotypes and domains of PH13 with GmCOP1b.

a. Diagram of protein structure of different haplotypes and domains of PH13. **b.** Auxotrophic assays showing the interactions between different haplotypes and domains of PH13 with GmCOP1b. Yeast cells transformed with indicated constructs were selected on -LW (lacking Leu and Trp) or -LWHA (lacking Leu, Trp, His and Ade) medium. **c.** β -galactosidase assays show the interaction strength of GmEID1 with each haplotype and domain of PH13. Data are means \pm SD ($n = 3$), p values were calculated by unpaired, two-tailed Student's t -tests.

2. Figure 4h, suppl. Figure 12d: The differences in the total protein amounts in different samples, reflected by the differences in HSP70 have not been included in the calculation. This is especially important since the HSP70 tends to be higher in NILH3 samples and low in the 24 h samples on the bottom panel of the suppl. Figure 12c.

Reply: We apologize for not providing the detailed information regarding the assay of protein abundance in our previous manuscript. Indeed, the levels of HSP70 protein have been included in the calculation of STF1/2 protein amounts in respective samples. We have now provided the detailed method for calculating the relative protein level in the Figure 4h legend (Line 841-847):

Fig. 4h: Comparison of STF1/2 protein abundance in NILs of PH13.

The relative expression level of STF1/2 proteins represented by REU (Relative Expression Unit) is calculated by the formula $[STF1/2] / [HSP70]$, in which ‘STF1/2’ and ‘HSP70’ indicated the digitized band intensity of STF1/2 or HSP70 in each sample collected at respective time point. The REU of STF1/2 in NIL^{H1} at ZT0 was arbitrarily set to 1. Data are shown as means \pm SD of three biological replications (Replication 1, Fig. 4g; Replication 2 and 3, Supplementary Fig. 12).

How many replicate experiments were performed for individual blots presented in the manuscript? I would suggest doing at least 2-3 replicates for each blot and then calculating the average before plotting the data (applicable to all the figures with immunoblots)

Reply: Thanks for your question. We have performed three independent replications for each immunoblot and calculating the average for all the plots which are shown in Fig. 4h and Supplementary Fig. 12, 14 and 18.

Supplementary Fig. 12: Biological replication 2 and 3 for Co-Immunoprecipitation (Co-IP) assay showing the interaction between each PH13 haplotype with GmCOP1b in tobacco leaves. Biological replication 1 is shown in Fig. 4f.

Supplementary Fig. 14: Biological replication 2 and 3 for STF1/2 abundance in NIL^{H1} and NIL^{H3}. Biological replication 1 is shown in Fig. 4g.

Supplementary Fig. 18: The additive effects of PH13 and PHP on STF1/2 abundance.
a and c, Three biological replications for immunoblots comparing the diurnal protein levels of STF1/2 in the *ph13-4* (a), *phd-1* mutants (c), and WT TL1^{H3}. The first trifoliolate leaves of 15-day-old seedlings grown under long-day condition in growth chamber were collected at 4-hour intervals. The membrane was probed with the anti-STF1/2 antibody, stripped, and then probed with the anti-HSP70 antibody. The asterisk indicates a non-specific band. **b and d,** The relative abundance of STF1/2 protein which was calculated by the formular as in Fig. 4h.

3. Nuclear localization of PH13: Suppl. Figure 5b shows the localization in the nucleus as well as in the cytoplasm. From the figure, they are distributed throughout and there are no dramatic differences in nuclear localization. This is especially peculiar since the nuclear localization of COP1 is necessary for its activity. I would suggest including these points in the discussion section

Reply: We greatly appreciate your valuable suggestion. We have included these points in the discussion section (lines 356-360) as follows:

“Furthermore, COP1 and SPAs have been reported to co-localize and function together within the nucleus in Arabidopsis⁴⁵⁻⁴⁶, whereas PH13 lacks an NLS (Nuclear Localization Signal) and is uniformly distributed in both the nucleus and cytoplasm (Supplementary Fig. 6b and 15). This suggests that PH13 may be involved in a distinct mechanism for regulating soybean growth and development.”

4. The TWAS RNA-seq data should be uploaded to a public database to give access to reviewers and readers

Reply: Thank you for your valuable reminder. We have now added detail description of the RNA-seq data (PRJCA014188) in the Data availability section (Line 602-606). The data can now be accessed at the following link: <https://ngdc.cncb.ac.cn/bioproject/browse/PRJCA014188>.

5. In Figure 1b, the legend does not provide any information about the green and orange colours used in the Manhattan plot. Also, apart from PH13, TWAS in Figure 1b shows at least two additional genes (orange colour)

Reply: We appreciate your valuable input. We have now added a description in the Fig. 1b legend (Line 774-775) to clarify that *“The green and orange dots are arranged in alternation to distinguish them from different chromosomes”*. We have also labeled all significant peaks of TWAS in Fig. 1b, and provided a description of the representing genes in Supplementary Table 3.

Fig. 1b: Identification of PH13 as a major QTL for plant height in soybean by TWAS.

Supplementary Table 3: List of plant height loci identified by TWAS.

Locus name	Gene ID	p.value	Description
PH04	Glyma.04G112700	1.60E-07	Encodes a WD40 repeat protein.
PH05	Glyma.05G246300	2.76E-10	Encodes an amidase, which belongs to one of the 36 carboxylate clamp (CC)-tetratricopeptide repeat (TPR) proteins.
PH11	Glyma.11G040900	8.00E-09	Encodes a Haloacid dehalogenase-like hydrolase.
PH13	Glyma.13G276700	9.51E-10	GmSPA3b, Encodes WD40 repeat protein involves a protein phosphorylation and protein kinase activity.
PH15	Glyma.15G000200	1.61E-07	NA
PH16.1	Glyma.16G091300	6.91E-08	GmAP1a, Encodes MADS box transcription factor.
PH16.2	Glyma.16G130000	1.75E-09	Encodes a HNRNP R-Like protein, which co-transcriptional Splicing of a key floral repressor gene FLOWERING LOCUS C (FLC).

6. qRT-PCR is not the right word. It should be RT-qPCR as it is not quantitative in real-time. This should be changed throughout the manuscript

Reply: We greatly appreciate your help to pointing out this error. We have revised “qRT-PCR” to “RT-qPCR” throughout the manuscript.

7. Suppl. Figure 12c: Do the asterisks indicate non-specific binding of the anti-STF antibody?

Reply: We appreciate your bring this to our attention. We have added “*The asterisk indicates a non-specific band*” in the legend of Supplementary Fig. 18 (previously labelled as Supplementary Fig. 12c) and Supplementary Fig. 14.

8. Data points in suppl. Figure 21d and e, suppl. Figure 22d and e: They can be shown scattered like the other plots in the same figure

Reply: Thanks for your suggestion. The data points representing branch number and node number have now been shown as scatter plots in Supplementary Fig. 28d and e (previously labelled as Supplementary Fig. 21d and e) and Supplementary Fig. 29d and e (previously labelled as Supplementary Fig. 22d and e).

Supplementary Fig. 28: Statistical analysis of agronomic traits of the *phd* mutants and WT TL1^{H3} under different planting density.

Supplementary Fig. 29: Comparison of the agronomic traits of the *phd* mutants, WT TL1^{H3}, and a local elite cultivar JY202.

9. Line 264: Please give reference for GmCRY1-qm mutant. Also, indicate how the *phd*-1/GmCRY1-qm line was made

Reply: We greatly appreciate your assistance in improving our manuscript. We have now added the information in the main text (Line 277-280):

“Next, we crossed the CRISPR-Cas9-engineered *GmCRY1s* quadruple (*GmCRY1s*-qm) mutant²¹ displaying constitutive ESE syndrome, with the *phd*-1 mutant carrying stocky phenotype, to obtain the *phd*-1/*GmCRY1s*-qm sextuple mutant (Supplementary Fig. 21a).”

10. Line 79: Change “of” to “in”

Reply: Revised.

11. Apart from PH13 and PHP, there are two more proteins GmSPA3c and GmSPA3d. The authors may include them in the alignment data shown in suppl. Figure 4

Reply: Thanks for your suggestion. We have added AtSPA3, GmSPA3c and GmSPA3d proteins in the alignment data (Supplementary Fig. 15).

Supplementary Fig. 15: Alignments of SPA3 family proteins in *Arabidopsis* and soybean.

The amino acid sequences of AtSPA3, PH13, PHP, GmSPA3c, and GmSPA3d were aligned using ClustalW Multiple alignment in MEGA7 and manually adjusted in GeneDOC software. Red boxes indicate the nuclear localization signals which were identified by the LOCALIZER program (<https://localizer.csiro.au/>).

Reviewer #4 (Remarks to the Author):

This manuscript starts out from a GWAS study for plant height in soybean, combined with TWAS for the same. A locus is identified from this study (PH13), which is then studied in further detail. The authors make the case that PH13 is homologous to Arabidopsis SPA genes, which work in a complex with COP1 towards protein degradation. The wrap up with showing that engineering PH13 affects soybean yield.

The manuscript is well written and has a logical flow. It does take a very similar path as a previous paper from some of the same authors published in Molecular Plant a few years ago, that studies components of the same regulatory pathway towards plant height in soybean.

In your manuscript you conclude that PH13 is a SPA-homologue, acting together with COP1 to regulate stability of STF1/2, which are HY5 homologues. This may be true, and confirms exactly what is very well established already in Arabidopsis. My concern is, therefore, that your study is rather confirmatory of Arabidopsis knowledge, now in soybean.

You do take this further to show that yield is affected, but this was shown previously already for soybean STF1/2 and CRY manipulation; essentially the pathway you study here as confirmed in your current data also.

Comment 1:

In your GWAS and TWAS you decided to focus on PH13 since this was the locus peak that overlapped between both. However, several other peaks showed up, some even with stronger scores. It would be relevant to explain more about their identity. A table with all significant peaks, their identity, gene family and function (if known) would be very helpful.

Reply: We appreciate your suggestion. Although *PH13* is the only locus showing overlap peak between GWAS and TWAS, it is possible that some other genes with significant signals may also play essential roles in plant height regulation. We have labeled these genes in Fig. 1b and added a table containing all significant peaks and their identity, as well as their potential functions (Supplementary Table 3). Please refer to our response to the question 5 of Reviewer #3.

Comment 2:

I am not sure that Figure 3 really has to be a main figure, it would do well as a supplemental figure.

Reply: We are deeply grateful for your assistance in improving the quality of our manuscript. Our research has revealed that the haplotype 3 of *PH13* has been subject to significant positive selection for soybean genetic improvement at high latitudes. In light of this discovery, we believe it would be beneficial to feature Figure 3 as a main figure to support this conclusion.

Comment 3:

You conclude, based on geographical distribution of the different PH13 alleles and knowledge on breeding patterns, that this gene is causally related to selection in soy breeding. It would be important for this claim to be supported by proof around this specific locus. For example, bringing the PH13H1 allele into a modern, dwarfed variety that carries the PH13H3 allele (with its

insertion), should restore plant height.

Causality: I am missing introgression of PH13H1 into PH13H1-expressing cultivars to confirm that really this locus is causal to the height and yield phenotypes of such cultivars.

Reply: We greatly appreciate the reviewer for bring this question to our attention, and apologize for any confusion caused by the imprecise use of terminology related to transgenic lines in the previous version of our manuscript. We have successfully transformed the *PH13^{H1}* gene into a dwarfed variety, TL1^{H3}, which carries the *PH13^{H3}* allele. Our results demonstrated that the *PH13^{H1}* allele was able to restore the dwarf phenotype of TL1^{H3} (Fig. 2 and Supplementary Fig. 8). To improve clarity, we have changed the name of transgenic lines “*H1-OEs*” to “*H1-OEs/TL1^{H3}*” in the revised manuscript.

Fig. 2: Genetic confirmation of *PH13* as a plant height regulator.

a, Gross photos of the indicated lines grown under natural field conditions in the summer of Beijing. The CRISPR/Cas9-engineered mutants *ph13-1* and *ph13-2* are in the W82^{H1} (carrying *PH13^{H1}*) background, and *ph13-4* and *ph13-6* are in the TL1^{H3} (carrying *PH13^{H3}*) background. The *PH13^{H1}* overexpression lines (H1-OE1/TL1^{H3} and H1-OE2/TL1^{H3}) are in the TL1^{H3} background. Scale bar, 20 cm. **b**, Plant height of the indicated lines as shown in (a). Data are mean \pm SD (n=10). **c**, Plant height of the near-isogenic lines (NILs) carrying homozygous H1 (NIL^{H1}) and homozygous H3 (NIL^{H3}) in Beijing field conditions. The NILs were derived from the hybrid combination between W82^{H1} and TL1^{H3} (Supplementary Fig. 10). Data are mean \pm SD (n > 20). Above *p* values were calculated by unpaired, two-tailed Student's t-tests.

Supplementary Fig. 8: Phenotypic analysis of the *ph13* mutants and *PH13^{HI}* overexpression lines at seedling stage.

a, Seedling photos of *ph13* mutants and WT plants grown under long day conditions (16 h light / 8 h dark) in phytotron. **b and c**, Plant height and flowering time of indicated lines as in (a). **d**, Epicotyl longitudinal sections of the WT W82^{HI} and *ph13-1* mutant at seedling stage. Scale bar, 0.5 mm. **e**, Xylem cell length (left panel, all the cells within each red rectangle were measured) and epicotyl diameter (right panel) of indicated lines as in (d). **f**, Immunoblot showing the abundance of PH13^{HI}-3×Flag protein in the overexpression lines. **g**, Seedling phenotypes of *PH13^{HI}* overexpression lines and WT TL1^{HI} grown under long day conditions. **h and i**, Plant height and flowering time of indicated lines as shown in (g). Statistic data are shown as mean values ± SD (n ≥ 5). *p* values were calculated by unpaired, two-tailed Student's *t*-tests.

Comment 3:

To corroborate your interpretation of PH13 being a SPA-like protein, it would be useful to complement a PH13H3 soybean cultivar with a functional SPA homologue from Arabidopsis. Likewise, you clone PH13H1 and express it in Arabidopsis *spa* mutants to verify if these are truly homologues.

Reply: Thanks for your suggestion. We have transformed the *PH13^{HI}-3×Flag* construct into the *Arabidopsis spa134* mutant. The results showed that ectopic expression of *PH13^{HI}* was able to restore the dwarf phenotype of the *spa134* mutant, supporting that PH13 is a homologous protein of SPAs (Supplementary Fig 2b-e). We have now added the information in the main text (Line 129-133): “We then transformed the 35S::*PH13-3×Flag* construct into the *spa134* mutant³⁴. The phenotypic results indicated that ectopic expression of PH13 can at least partially rescued the dwarf phenotype of the *spa134* mutant at both seedling and adult vegetative stage, supporting that PH13 is homologous to the SPA family proteins (Supplementary Fig. 2b-e)”

Supplementary Fig. 2: Ectopic expression of *PH13* in the *Arabidopsis spa134* mutant.

a, Phylogenetic analysis of SPA family members from *Arabidopsis thaliana*, rice, and soybean. The tree was constructed using the Maximum Likelihood method in the MEGA7 software. **b**, Immunoblot analysis of PH13-3×Flag protein expression in the indicated lines using the anti-Flag antibody. HSP70 protein was used as a loading control. **c**, Comparison of hypocotyl lengths of indicated lines. Seedlings were grown for 4 days under blue light ($50 \mu\text{mol m}^{-2} \text{s}^{-1}$), red light ($5 \mu\text{mol m}^{-2} \text{s}^{-1}$), or darkness. Scale bar = 0.5 cm. **d**, Statistical analysis of hypocotyl lengths in (c). Data are means \pm SD ($n = 15$). Lowercase letters above the dots indicate significant differences ($p < 0.01$, ANOVA with Tukey's post-test). **e**, Visual phenotype of indicated plants grown under long day conditions for three weeks. Scale bar = 2 cm.

Comment 4:

In Fig 6 you draw root systems, but you have not looked into these. Therefore, any suggestion about root development should be avoided. This is quite relevant since HY5 and presumably STF1/2 in soybean, might control root architecture.

Reply: We thank the reviewer for bring this to our attention. We have conducted additional observation and found that the *phd* mutant exhibit a larger root system compared to the wild type (WT) at both seedling and maturity stages, as shown in Supplementary Fig. 26. We have included this information in the revised manuscript (Line 301-303):

“In addition, we observed that the *phd* mutants exhibit a larger root system at seedling stage in growth chamber and at maturity stage in Beijing field conditions (Supplementary Fig. 26).”

Supplementary Fig. 26: Effects of the *phd* mutation on root architecture of soybean.

a, Seedling phenotypes of *phd-1* mutants and WT TL1^{H3} grown under long day conditions (16 h light/8 h dark) in phytotron. The red arrow represents the position of the longest lateral root. Scale bar, 2 cm. **b-e**, Statistical analysis of the length of main root (**b**), number of lateral root (**c**), length of lateral root (**d**), and dry weight of root (**e**) of the indicated lines as in (**a**). Data are mean \pm SD ($n = 15$) with p values calculated by unpaired, two-tailed Student's t -tests. **f**, Representative images of the indicated lines grown under Beijing field conditions in 2021. Scale bar = 20 cm.

Comment 5:

Lines 270-272: You did not study shade tolerance, and can therefore also not draw the conclusion on "excellent shade-tolerance traits".

Reply: Thank you for bring this to our attention. We apologize for the overstatement and have adjusted the statement in the main text as follows (Line 287-289):

*"Collectively, these results demonstrate that the *phd* mutant is characterized by ideal shade-resistant traits, which may be advantageous for high-density planting or intercropping at high latitudes".*

** See Nature Portfolio's author and referees' website at www.nature.com/authors for information about policies, services and author benefits.

Reviewers' Comments:

Reviewer #1:

Remarks to the Author:

I am satisfied with the edits made and have no remaining concerns. I recommend publication of this work.

Reviewer #2:

Remarks to the Author:

The authors have fully addressed my comments from the previous review.

Reviewer #3:

Remarks to the Author:

The authors have addressed all the major and minor comments satisfactorily. However, I still have some very minor corrections to suggest:

1. Line 843: Please correct the typo and change "formular" to "formula".
2. Suppl. Figure 6: Please use the standard abbreviation "RT-qPCR" instead of "quantitative RT-PCR", as this is the more accurate and widely accepted term for reverse transcription-quantitative polymerase chain reaction.
3. Suppl. Figure 12 and Suppl. Figure 18: Please change "replication" to "replicate", both within the figures and in the description, to avoid confusion with the biological process of replication.

Reviewer #4:

Remarks to the Author:

You have greatly improved your manuscript. All my previous questions have been addressed very well, sometimes by additional data, sometimes by carefully explaining specific aspects.

I am also impressed with your new data on root growth, which I had not anticipated. If you would have recorded data on lateral root number and/or lateral root density (number of lateral roots per unit main root length), it would be very useful to add those.

I have no further comments or requests and I believe this manuscript can make an important contribution in the field of plant density responses and shade avoidance.

Reply to referees:

We appreciate the encouragement provided by the four reviewers and the editor. We have carefully revised our manuscript in response to valuable comments and editorial requests. Please see our point-by-point response below.

#####

REVIEWER COMMENTS

Reviewer #1 (Remarks to the Author):

I am satisfied with the edits made and have no remaining concerns. I recommend publication of this work.

Reviewer #2 (Remarks to the Author):

The authors have fully addressed my comments from the previous review.

Reviewer #3 (Remarks to the Author):

The authors have addressed all the major and minor comments satisfactorily. However, I still have some very minor corrections to suggest:

1. Line 843: Please correct the typo and change "formular" to "formula".

Reply: We appreciate you so much for the careful review. We have changed the typo "formular" to "formula".

2. Suppl. Figure 6: Please use the standard abbreviation "RT-qPCR" instead of "quantitative RT-PCR", as this is the more accurate and widely accepted term for reverse transcription-quantitative polymerase chain reaction.

Reply: Thanks for your suggestion. we have used the standard abbreviation "RT-qPCR" instead of "quantitative RT-PCR".

3. Suppl. Figure 12 and Suppl. Figure 18: Please change "replication" to "replicate", both within the figures and in the description, to avoid confusion with the biological process of replication.

Reply: Thanks for your suggestion. We have changed "replication" to "replicate".

Reviewer #4 (Remarks to the Author):

You have greatly improved your manuscript. All my previous questions have been addressed very well, sometimes by additional data, sometimes by carefully explaining specific aspects.

I am also impressed with your new data on root growth, which I had not anticipated. If you would have recorded data on lateral root number and/or lateral root density (number of lateral roots per unit main root length), it would be very useful to add those.

Reply: Thank you for your valuable suggestion. We will conduct a more thorough investigation of the root phenotype as you have recommended.

I have no further comments or requests and I believe this manuscript can make an important contribution in the field of plant density responses and shade avoidance.

Response to general comments:

Thank you very much for your positive and constructive review, which greatly improved our manuscript. We are pleased that our revisions were able to address all your concerns and comments to your satisfaction.